# A stress-induced cilium-to-PML-NB route drives senescence initiation

Xiaoyu Ma[1], Yingyi Zhang[1], Yuanyuan Zhang[2], Xu Zhang [3,4], Yan Huang[1], Kai He[1], Chuan Chen[1], Jielu Hao[1], Debiao Zhao[1], Nathan K. LeBrasseur [3,4,5], James L. Kirkland [3], Eduardo N. Chini [3,6], Qing Wei [7] ✉, Kun Ling [1] ✉ & Jinghua Hu [1,8,9] ✉

Cellular senescence contributes to tissue homeostasis and age-related pathologies. However, how senescence is initiated in stressed cells remains vague. Here, we discover that exposure to irradiation, oxidative or inflammatory stressors induces transient biogenesis of primary cilia, which are then used by stressed cells to communicate with the promyelocytic leukemia nuclear bodies (PML-NBs) to initiate senescence responses in human cells. Mechanistically, a ciliary ARL13B-ARL3 GTPase cascade negatively regulates the association of transition fiber protein FBF1 and SUMO-conjugating enzyme UBC9. Irreparable stresses downregulate the ciliary ARLs and release UBC9 to SUMOylate FBF1 at the ciliary base. SUMOylated FBF1 then translocates to PML-NBs to promote PML-NB biogenesis and PML-NB-dependent senescence initiation. Remarkably, *Fbf1* ablation effectively subdues global senescence burden and prevents associated health decline in irradiation-treated mice. Collectively, our findings assign the primary cilium a key role in senescence induction in mammalian cells and, also, a promising target in future senotherapy strategies.

Cellular senescence is a programmed response that healthy cells enter an irreversible growth-arrest and apoptosis-resistant state, established by constitutive activation of the p16[INK4a]-RB and/or p53-p21[CIP1] pathways, when experiencing irreparable stresses[1,2]. Senescent cells secret excessive pro-inflammatory cytokines, chemokines, proteases, and growth factors, which are collectively known as the senescence-associated secretory phenotype (SASP)[1]. Senescence occurs throughout life and can be beneficial in tissue repair and homeostasis[3,4]. However, unchecked senescence results in deleterious SASP secretion that contributes to increased cancer risk and age-related diseases[3]. Genetic or pharmacological clearance of senescent cells effectively

improves survival and/or healthspan in rodent models[5]. As such, targeting senescence has emerged as a promising therapeutic strategy to prevent or treat aging comorbidities and cancer.

The promyelocytic leukemia nuclear bodies (PML-NBs) are proteinaceous nuclear structures with instrumental roles in regulation of various stress-induced responses including senescence[6,7]. It has been commonly accepted that PML-NBs serve as a central hub of stress-induced responses. The structural basis of PML-NBs is the PML protein, a tumor suppressor dynamically and functionally associated with hundreds of PML-NB components (also termed PML interactome)[8]. In response to stresses, the number, composition, size, and

[1]Department of Biochemistry and Molecular Biology, Mayo Clinic, Rochester, MN, USA. [2]Department of Clinical Genetics, ShengJing Hospital of China Medical University, Shenyang, Liaoning, China. [3]Mayo Clinic Robert and Arlene Kogod Center on Aging, Mayo Clinic, Rochester, MN, USA. [4]Department of Physical Medicine and Rehabilitation, Mayo Clinic, Rochester, MN, USA. [5]Department of Physiology and Biomedical Engineering, Mayo Clinic, Rochester, MN, USA. [6]Department of Anesthesiology, Mayo Clinic, Jacksonville, FL, USA. [7]Center for Energy Metabolism and Reproduction, Institute of Biomedicine and Biotechnology, Shenzhen Institutes of Advanced Technology, Chinese Academy of Sciences (CAS), Shenzhen, China. [8]Mayo Clinic Robert M. and Billie Kelley Pirnie Translational Polycystic Kidney Disease Center, Mayo Clinic, Rochester, MN, USA. [9]Division of Nephrology and Hypertension, Mayo Clinic, Rochester, MN, USA. ✉e-mail: qing.wei@siat.ac.cn; ling.kun@mayo.edu; hu.jinghua@mayo.edu

posttranslational modifications (e.g., SUMOylation, phosphorylation, and acetylation) of PML-NBs undergo significant changes[8]. However, molecular insights into stress-induced biogenesis and function heterogeneity of PML-NBs remain poorly understood.

Primary cilia evolve as sensory devices that protrude from most eukaryotic cell surfaces to convert environmental cues into diverse cellular signaling that govern proliferation, differentiation, and tissue homeostasis[9,10]. So far, at least 35 human disorders, such as Joubert syndrome (JBTS), Bardet-Biedl syndrome (BBS), and nephronophthisis (NPHP), have been characterized as ciliopathies[11,12]. Ciliopathies occur as syndromic disorders that collectively affect most organs/tissues in human body. Despite their importance, how primary cilia sense and transduce signals into intracellular responses remain poorly defined. Of note, increased ciliation was observed in senescent mammalian cells[13–15], and intrinsic genotoxic stress can also induce ciliogenesis[13]. It is thus conceivable that cells may use primary cilia to sense cues and regulate stress-induced responses.

In this study, we discovered that irradiation (IR) induces robust but transient ciliogenesis in stressed cells. Further analyses revealed that transient ciliogenesis in stressed mammalian cells are accompanied by acute downregulation of two ciliary GTPases ARL13B and ARL3, and the opposite upregulation of a transient fiber (TF) component FBF1. ARL13B and ARL3 are mutated in Joubert syndrome[16,17]. ARL13B serves as a guanine nucleotide exchange factor (GEF) to activate ARL3[18,19]. Mechanistically, we discovered that the ARL13B-ARL3 GTPase cascade negatively regulates UBC9-mediated SUMOylation of FBF1 at the ciliary base. Irreparable stresses suppress ARLs expression and thus enhance FBF1 SUMOylation, which is required for FBF1 translocation from the ciliary base to PML-NBs to promote stress-induced PML-NB upregulation and initiates senescence. Excitingly, genetic ablation of *Fbf1* protects mice from IR-induced senescence propagation and associated frailty. Our study thus highlights a promising strategy to target primary cilia in future therapeutics to counteract senescence-associated pathologies.

## Results

### Transient ciliogenesis is required for senescence initiation in stressed human cells

When studying irradiation (IR)-induced senescence in human fetal lung fibroblasts (IMR-90) cells[20,21], we observed transient but robust ciliogenesis, as evident by both ciliation ratio and cilia length, during the progression of senescence (Fig. 1a). Consistently, transient ciliogenesis was also observed in oxidative ($H_2O_2$) or inflammatory (IL1 β) stressor-induced senescence in IMR90 cells (Supplementary Fig. 1a, b). To determine if the primary cilium is required for senescence induction, we knocked down *KIF3A* or *IFT88*, which encodes essential ciliogenesis regulators[22–24]. In *KIF3A* or *IFT88* deficient cells, transient ciliogenesis was disrupted as expected in IR-, $H_2O_2$-, or IL1β-treated IMR-90 cells (Supplementary Fig. 1c–f). Remarkably, *siKIF3A* or *siIFT88* treatment abolishes senescence-associated β-galactosidase (SA-β-gal) staining and senescence responses (p16[INK4A], p21[CIP1]) in stressed IMR-90 cells (Supplementary Fig. 1g–i). These results support a central role for primary cilia in regulating senescence induced by different stressors. We further analyzed the expression of dozens of proteins required for cilia formation and/or function in IR-treated IMR-90 cells, including IFT proteins (IFT74, IFT88, IFT140, IFT144), transition zone components (MKS1, MKS3, MKS5, NPHP1, NPHP4), transition fibers components (FBF1, CEP164, CEP83), BBS proteins (BBS1, BBS5, BBS12), and the ciliary GTPases (ARL3, ARL6, ARL13B, RAB11) (Fig. 1b). Remarkably, we found that, during senescence induction, ARL3 and ARL13B GTPases mutated in Joubert syndrome[16,17] were gradually downregulated, whereas transition fiber component FBF1[25] was steadily upregulated (Fig. 1b). We reasoned that ARL3/ARL13B/FBF1 might be key players involved in cilia-mediated senescence since no changes were detected for other ciliary proteins examined. Intriguingly, the moment that the

level of ARLs and FBF1 change during senescence progression occurs simultaneously after cilia form in stressed cells (Fig. 1a, b). These data suggest an opposite regulation of the ciliary ARLs and TF component FBF1 during senescence induction in stressed cells.

### ARL3 deficiency promotes senescence in stressed human cells

To confirm that ARL GTPases indeed regulate cellular senescence, we used two primary cell models commonly used in senescence studies, human IMR-90 cells and mouse embryonic fibroblasts (MEFs). Compared to IR-treated controls, IR-treated *shARL3* IMR-90 cells exhibited significantly increased activity of SA-β-Gal (Fig. 1c), elevated expression of molecular senescence markers (Fig. 1d) and SASP genes (Fig. 1e). Similarly, *Arl3⁻/⁻* MEFs exhibited upregulated senescence responses than control cells after IR treatment (Supplementary Fig. 2a–d). Meanwhile, we established CRISPR/Cas9-engineered *ARL3⁻/⁻* cells to confirm observation obtained from RNAi studies. Primary cells like IMR-90 are notorious for the difficulty in genetic manipulations and clone selection due to their limited population doublings. We thus used human renal collecting tubule epithelia (RCTE) cell line for genome engineering. *ARL3⁻/⁻* RCTE cells showed upregulated SA-β-Gal activity, increased expression of molecular senescence markers and SASP genes (Supplementary Fig. 2e–g), and lower death (Supplementary Fig. 2f, h) when compared with wildtype (WT) parental cells after IR treatment. Intriguingly, the protein expression of ARL3 is strongly inhibited upon senescence induction in all mammalian cells tested (Fig. 1d and Supplementary Fig. 2c, f). Collectively, our data demonstrate that ARL3 acts as a negative regulator of senescence in mammalian cells.

### ARL13B recapitulates the role of ARL3 in senescence regulation

As with ARL3, *ARL13B*-knockdown IMR-90 cells (Fig. 1f–h), or *ARL13B⁻/⁻* RCTE cells (Supplementary Fig. 2i–k), show stronger senescence responses than control cells after IR treatment. Combined with the fact that ARL13B and ARL3 are the only two ciliary proteins downregulated in senescence and that ARL13B acts as a GEF to activate ARL3[18,19], we hypothesize that ARL13B and ARL3 may act in the same pathway to regulate senescence.

### FBF1 promotes cellular senescence in stressed human cells

In accord with the drastic upregulation of FBF1 in senescent cells, FBF1 deficiency suppressed senescence responses (Fig. 2a–c), increased apoptosis (Fig. 2d) and reduced viability (Fig. 2e) in IR-treated s*hFBF1* IMR-90 cells. Studies in MEFs isolated from *Fbf1^tm1a/tm1a* mutant mice[26] further supported that FBF1 deficiency led to reduced expression of senescence markers and SASP genes (Supplementary Fig. 3a, b) and reduced viability (Supplementary Fig. 3c) after IR treatment. As what we did for ARLs, we then established CRSPR/Cas9-engineered *FBF1⁻/⁻* RCTE cell lines. Consistently, *FBF1⁻/⁻* RCTE cells recapitulated the phenotypes of s*hFBF1* IMR-90 or *Fbf1^tm1a/tm1a* MEF cells in IR-induced senescence (Supplementary Fig. 3d–f). As expected, FBF1 deficiency also blocked $H_2O_2$- or IL1β-induced senescence responses in IMR-90 cells (Supplementary Fig. 3g–i). Remarkably, overexpression of FBF1 alone in IMR-90 cells strongly promoted senescence responses in stressed cells (Fig. 2f–h). Together, these results highlight that FBF1 plays a decisive role in driving senescence initiation in mammalian cells.

### FBF1 translocates from primary cilia to PML-NBs upon senescence induction

It is known that another TF component, CEP164, regulates DNA damage responses by translocating to the nuclear DNA damage foci in IR-treated human retinal epithelial cells[27]. This intrigued us to study where FBF1 regulates senescence. We then set out to examine the subcellular localization of FBF1 in senescent cells. Intriguingly, although the ciliary localization of FBF1 shows no change,

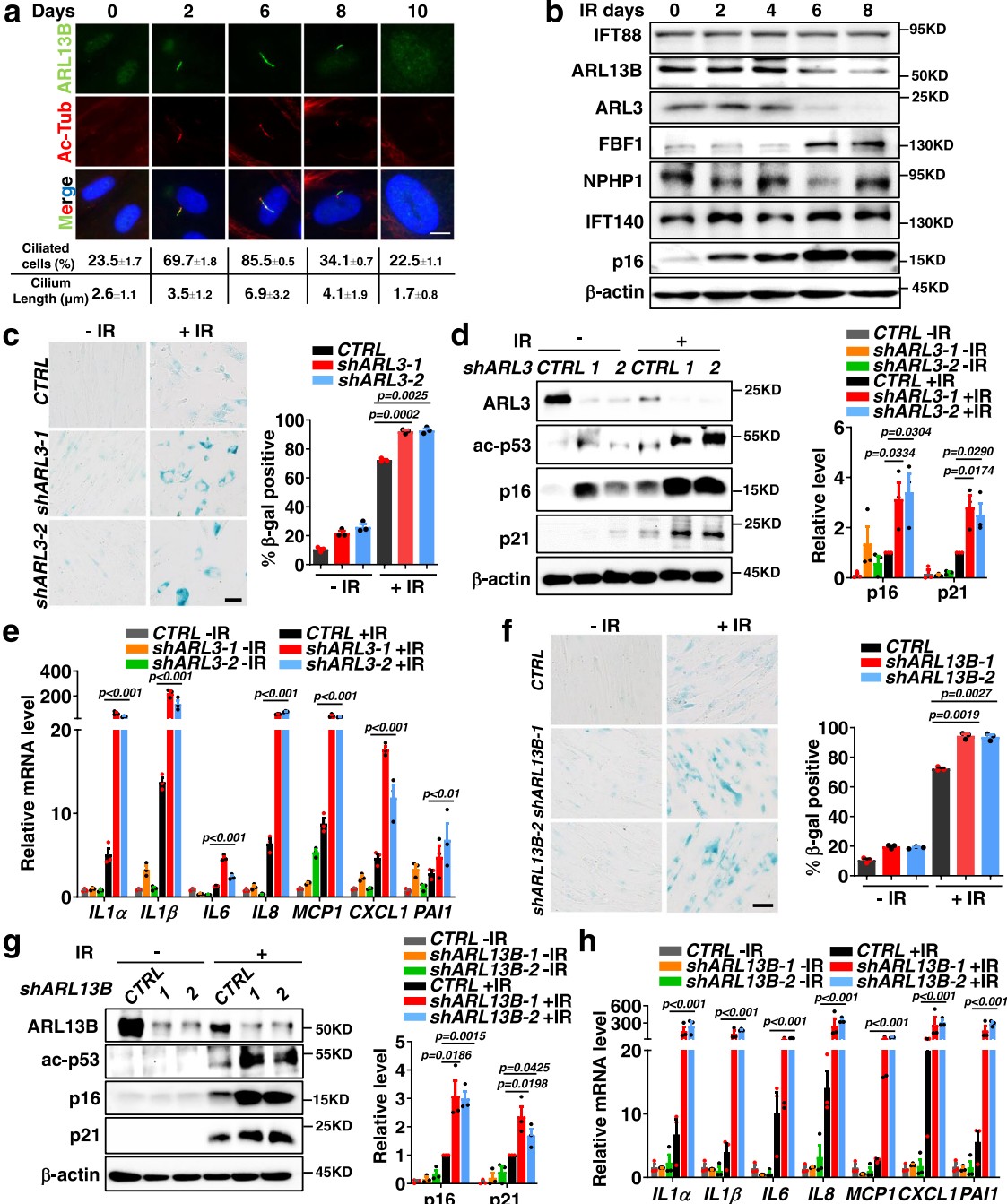

**Fig. 1 | Deficiency of Joubert syndrome GTPase ARL3 or ARL13B promotes DNA damage-induced senescence. a** Immunofluorescent images for changes of primary cilia in IMR-90 cells after senescence-induction by irradiation. Cilia were labeled with acetylated tubulin. Scale bar, 10 μm. Three experiments were repeated independently with similar results. **b** Western blot showing changes in ciliary protein levels after senescence-induction in IMR-90 cells by irradiation. Three experiments were repeated independently with similar results. **c**–**e** SA-β-gal staining (n>100 cells per experiment) (**c**), western blot of senescence markers (**d**) and relative mRNA levels of SASP genes (**e**) in IMR-90 cells stably expressing control shRNA or sh*ARL3* at day 10 post-irradiation. **f**–**h** SA-β-gal staining (**f**), western blot of senescence markers (**g**) and relative mRNA levels of SASP genes (**h**) in IMR-90 cells stably expressing control shRNA or sh*ARL13B* at day 10 post-irradiation. Scale bar, 200 μm. Results (**c**–**h**) from n=3 independent experiments were statistically analyzed and plotted as means ± SEM. Brown-Forsythe and Welch ANOVA tests was used for **c** and **f**. Two-tailed Student's unpaired *t*-test was used for analysis in **d** and **g**. Two-way ANOVA followed by Bonferroni multiple-comparison analysis was used for **e** and **h**. Source data are provided as a Source Data file.

endogenous FBF1 gradually translocated into the nucleus after IR treatment and colocalized with PML-NBs (Fig. 3a), a group of highly dynamic, proteinaceous structures with validated function linked to stress-induced responses[28]. The nuclear translocation of FBF1 occurred continually for a few days in stressed cells, accompanied by increased PML-NB numbers. The reported nuclear foci of CEP164 are not PML-NBs[27], suggesting FBF1 and CEP164 act in different pathways

to regulate stress responses. PML-NB upregulation is well-studied hallmark for senescent cells. Super-resolution structured illumination microscopy (SIM) analysis confirmed that translocated FBF1 indeed colocalized with PML, a core component of PML-NB (Fig. 3b). Examination of isolated nuclear and cytoplasmic proteins confirmed that FBF1 gradually entered the nucleus upon IR treatment (Supplementary Fig. 4a). Strikingly, FBF1 deficiency completely suppressed

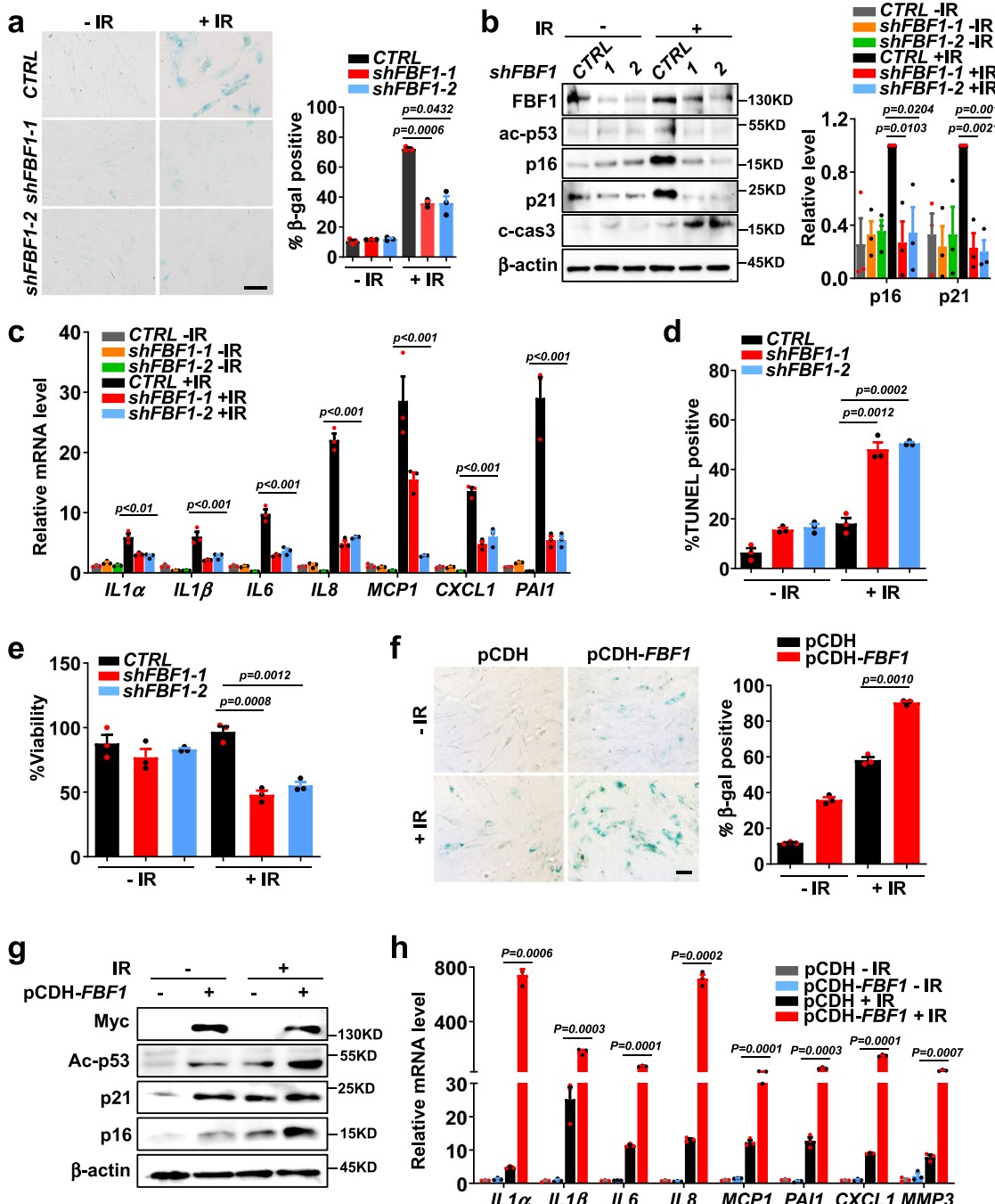

**Fig. 2 | Transition fiber component FBF1 is a key player in DNA damage-induced senescence. a**–**c** SA-β-gal staining (n>100 cells per experiment) (**a**), western blot of senescence markers (**b**) and relative mRNA levels of SASP genes (**c**) in IMR-90 cells stably expressing control shRNA or sh*FBF1* at day 10 post-irradiation. Scale bar, 200 μm. **d**, **e** TUNEL assay (n=30 cells per experiment) (**d**) and viability assay (**e**) in IMR-90 cells stably expressing control shRNA or sh*FBF1* at day 3 post-irradiation. **f**–**h**, SA-β-gal staining (**f**), western blot of senescence markers (**g**) and relative mRNA levels of SASP genes (**h**) in IMR-90 cells overexpressed with plasmid pCDH vector or pCDH-FBF1-Myc at day 7 after irradiation. Scale bar, 200 μm. All results from $n = 3$ independent experiments were statistically analyzed and plotted as means ± SEM. Brown-Forsythe and Welch ANOVA tests was used for **a** and **f**. Two-way ANOVA followed by Bonferroni multiple-comparison analysis was used for **c**. Two-tailed Student's unpaired t-test was used for analysis in **b**, **d**, **e** and **h**. Source data are provided as a Source Data file.

IR-induced PML-NB upregulation in IMR-90 cells (Fig. 3c) or RCTE cells (Supplementary Fig. 4b). Similar observations were made in $H_2O_2$- or IL1β-induced senescence (Supplementary Fig. 4c, d). In contrast, FBF1 overexpression significantly upregulated PML-NBs in stressed cells (Supplementary Fig. 4e). In agreement with the critical role of cilia in senescence regulation, *siKIF3A* or *siIFT88* treatment suppressed PML-NB translocation of FBF1 in IR-, $H_2O_2$- or IL1β-treated IMR-90 cells (Supplementary Fig. 4f–h).

## Deficiency of ARL3 or ARL13B promotes IR-induced PML-NB translocation of FBF1

The intriguing fact that three ciliary proteins, ARL3, ARL13B and FBF1, act as either negative or positive regulators of senescence suggests that the primary cilium might be an overlooked organelle critical for senescence regulation. Deficiency of *ARL3* or *ARL13B* strongly promoted PML-NB translocation of FBF1 and PML-NB upregulation in IR-treated IMR-90 or RCTE cells (Fig. 3d and Supplementary Fig. 5a),

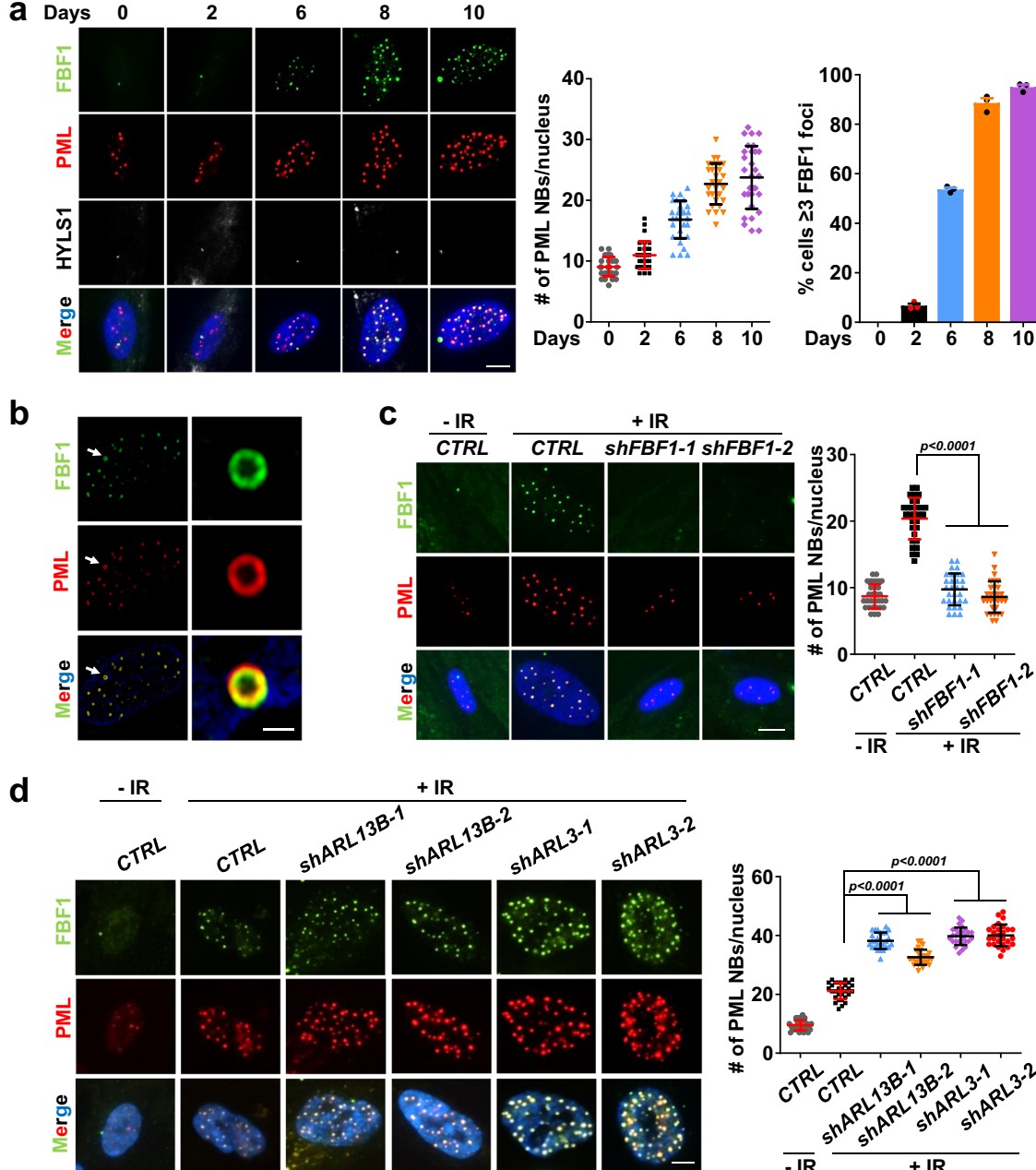

**Fig. 3 | ARL13B and ARL3 negatively regulate DNA damage-induced PML-NB translocation of FBF1, a prerequisite for stress-induced PML-NB upregulation.**
**a** Immunofluorescent images in IMR-90 cells after irradiation using antibodies against FBF1 and PML. Basal bodies were labeled with HYLS1. Scale bar, 10 μm. Number of PML NBs per nucleus (*n* = 30 cells) and percentage of cells containing ≥3 FBF1 foci after irradiation were quantified. **b** Structured illumination microscopic images of the nucleus of a senescent IMR-90 cell stained for FBF1 and PML. Scale bar, 2 μm. Three experiments were repeated independently with similar results.

**c** Localization of FBF1 and PML in control or *FBF1*-knockdown IMR-90 cells at day 10 after irradiation. n=30 cells. Scale bar, 10 μm. **d** Localization of FBF1 and PML in control, *ARL3*-knockdown, or *ARL13B*-knockdown IMR-90 cells at day 10 after irradiation. *n* = 27 cells. Scale bar, 10 μm. All results from *n* = 3 independent experiments were statistically analyzed and plotted as means ± SEM. One-way ANOVA followed by Bonferroni multiple-comparison analysis was employed for **c** and **d**. Source data are provided as a Source Data file.

suggesting that the ciliary ARLs and FBF1 functionally converge through a PML-NB-dependent manner to regulate senescence. Moreover, cilia suppression through *KIF3A* knockdown inhibited the impacts of ARL13B deficiency on IR-induced PML-NB biogenesis, PML-NB translocation of FBF1, and senescence responses (Supplementary Fig. 5b–d), supporting that the function of the ARL-FBF1-PML-NB axis in senescence regulation is cilia-dependent. To further confirm if the ARL-FBF1-axis-regulated senescence is conserved in other cell types, we examined skin primary fibroblast BJ cells. As shown in Supplementary Fig. 6a and b, upon IR treatment, *shARL13B* or *shARL3* BJ cells

exhibited significantly upregulated senescence responses. Consistently, deficiency of ARL3 or ARL13B strongly promoted PML-NB translocation of FBF1 and PML-NB biogenesis in IR-treated BJ cells (Supplementary Fig. 6d). FBF1 deficiency suppressed senescence responses and PML-NB biogenesis in IR-treated BJ cells (Supplementary Fig. 6a, c and e).

## IR induces FBF1-UBC9 association and FBF1 SUMOylation

A distinct feature of PML-NBs is the recruiting of SUMOylated proteins[8]. We previously discovered that UBC9, the sole E2 SUMO-

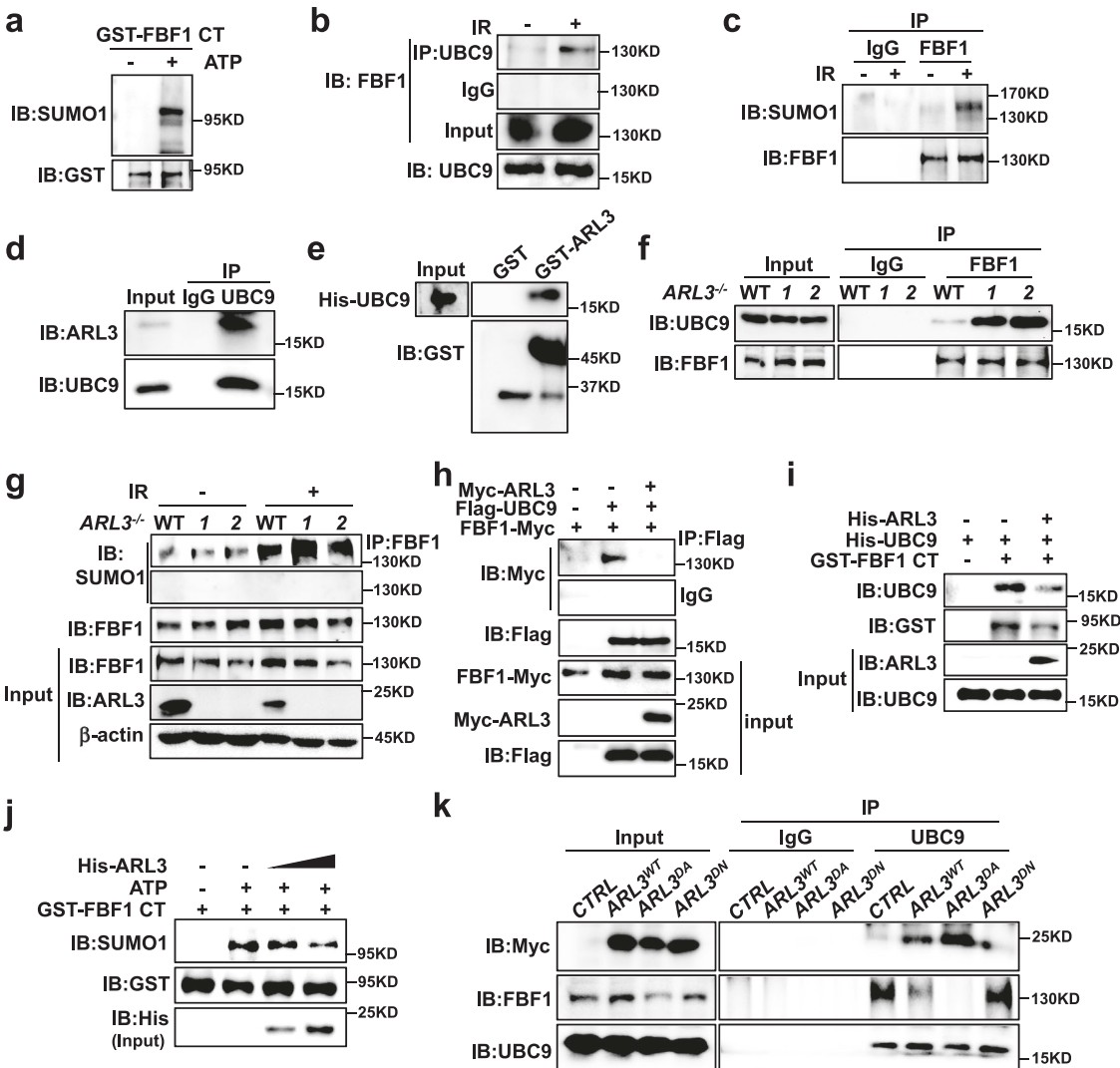

**Fig. 4 | DNA damage induces FBF1-UBC9 association and FBF1 SUMOylation, which are negatively regulated by the ARL13B-ARL3 GTPase cascade. a** In vitro SUMOylation assay of FBF1 and western blot using a SUMO1 antibody. GST tagged 598–1148 domain FBF1 was purified using *E. coli*. **b** Endogenous UBC9 immuno-precipitates with FBF1 in RCTE cells at day 7 after irradiation. **c** Endogenous FBF1 immunoprecipitates with SUMO1 in RCTE cells at day 7 after irradiation. **d** Endogenous UBC9 immunoprecipitates with ARL3 in RCTE cells. **e** GST pull-down assay shows the ARL3 and UBC9 interaction. **f** Endogenous FBF1 immunoprecipi-tates with UBC9 in WT or *ARL3⁻/⁻* RCTE cells at day 7 after irradiation. **g** Endogenous FBF1 immunoprecipitates with SUMO1 in WT or *ARL3⁻/⁻* RCTE cells with or without IR

treatment. **h** ARL3 inhibited the interaction between FBF1 and UBC9 by co-immunoprecipitation in 293T cells. **i** ARL3 reduced the interaction between FBF1 and UBC9 by GST pull-down assay. GST-tagged C-terminal fragment (598-1148) of FBF1, His-ARL3 and His-UBC9 were purified using *E. coli*. **j** In vitro SUMOylation assay was performed in a reaction with GST-tagged C-terminal fragment (598-1148) of FBF1 and His-ARL3 purified in *E. coli*. Western blot was detected by SUMO1, GST and His antibody. **k** Endogenous UBC9 immunoprecipitates with Myc-ARL3 or FBF1 in ARL3^WT, ARL3^DA or ARL3^DN overexpression RCTE cells at day 7 after irradiation. Three experiments were repeated independently with similar results (**a–k**). Source data are provided as a Source Data file.

conjugating enzyme, associates with and SUMOylates ARL13B in the cilia of renal epithelial cells[29]. However, we did not detect ARL13B or ARL3 translocation into the nucleus in IR-treated cells. Since FBF1 translocates to PML-NBs in senescent cells, we examined if FBF1 could be SUMOylated. in vitro SUMOylation assay detected strong SUMOy-lation in the FBF1 C-terminal fragment (Fig. 4a). Immunoprecipitation (IP) assay further confirmed endogenous association between FBF1 and UBC9 and FBF1 SUMOylation, which increased greatly upon IR treatment (Fig. 4b, c and Supplementary Fig. 7a).

### The ARL13B-ARL3 GTPase cascade inhibits UBC9-mediated FBF1 SUMOylation

We then determined how the ciliary ARLs contributes to IR-induced FBF1 SUMOylation. Intriguingly, endogenous ARL3 was associated with UBC9 in non-senescent cells (Fig. 4d). Purified GST-ARL3 interacted with His-UBC9 in vitro (Fig. 4e). *ARL3* deficiency significantly promoted

FBF1-UBC9 interaction and IR-induced FBF1 SUMOylation (Fig. 4f, g), whereas ARL3 overexpression impaired the FBF1-UBC9 interaction (Fig. 4h). Purified His-ARL3 compromised the FBF1-UBC9 interaction (Fig. 4i) and reduced FBF1 SUMOylation in a dose-dependent man-ner (Fig. 4j), suggesting that ARL3 directly competes with FBF1 for UBC9 binding. The downregulation of ARL3 upon IR treatment thus releases UBC9 to SUMOylate FBF1.

Both ARL13B and ARL3 act as negative senescence regulators. Our previous discovery that ARL13B serves as a GEF to activate ARL3 suggests that ARL3 regulation of FBF1 SUMOylation may depend on its GTPase activity. To test this hypothesis, we overexpressed WT, GTP-lock (dominant active, DA) or GDP-lock (dominant negative, DN) variants of ARL3 in RCTE cells. In agreement with our hypothesis, the ARL3^DA variant showed stronger interaction with UBC9 than ARL3^WT, whereas the ARL3^DN variant exhibited the opposite phenotype (Fig. 4k and Supplementary Fig. 7b). As expected, the FBF1-UBC9 interaction

was largely attenuated in ARL3[DA]-expressing cells (Fig. 4k). Similar to the results in *ARL3* knockdown cells, *ARL13B* deficiency led to increased FBF1-UBC9 association (Supplementary Fig. 7c), whereas overexpression of ARL13B[WT], but not ARL13B[DN], impairs IR-induced FBF1-UBC9 association (Supplementary Fig. 7d). Consistently, ARL3[DA]-overexpressing cells were more resistant to senescence induction in IR-treated IMR-90 cells (Supplementary Fig. 7e–g).

## SUMO protease SENP1 counteracts FBF1 SUMOylation and PML-NB translocation

Like most posttranslational modifications, SUMOylation is a reversible process[30]. Sentrin/SUMO-specific proteases (SENPs) catalyze the deconjugation of SUMO[31]. Notably, endogenous or overexpressed SENP1 strongly labeled the ciliary base in non-senescent cells (Fig. 5a and Supplementary Fig. 8a). SIM studies further showed that endogenous SENP1 labeled a specific region that overlaps with FBF1 on transition fibers in non-senescent cells (Fig. 5b). FBF1 could coimmunoprecipitate either overexpressed or endogenous SENP1 (Fig. 5c, d), indicative of a direct association between FBF1 and deSUMOylase SENP1. Intriguingly, IR treatment greatly reduced the ciliary SENP1 signal and the total SENP1 level, and disrupted FBF1-SENP1 association (Fig. 5e, f). As expected, knockdown of *SENP1* drastically promoted PML-NB translocation of FBF1 and PML-NB upregulation in IR-treated cells (Supplementary Fig. 8b, c). Overexpression of WT SENP1, but not a SENP1 variant that lost enzymatic activity[32], diminished UBC9-mediated FBF1 SUMOylation (Fig. 5g). Collectively, these results reveal a novel role for SUMO protease SENP1 in negatively regulating senescence by counteracting FBF1 SUMOylation at the ciliary base.

To further confirm that FBF1 regulation by SENP1 is cilia-dependent, we used a chemically inducible cilia trapping system[33]. Trapping SENP1 to the ciliary base could effectively attenuate both PML-NB translocation of FBF1 and PML-NB upregulation in IR-treated *ARL3*-deficient cells (Fig. 5h and Supplementary Fig. 8d). These data strongly suggest that removal of the ciliary SENP1 in stressed cells is crucial for ARL-regulated FBF1 SUMOylation, PML-NB translocation of FBF1, and senescence initiation.

## TF localization is a prerequisite for PML-NB translocation of FBF1 in stressed cells

To further distinguish if it is the ciliary or non-ciliary FBF1 responsible for initiating senescence responses, we knocked down the key TF structural components CEP83 or SCLT1. *CEP83* or *SCLT1* deficiency disrupts TF formation[34]. Knockdown of either *CEP83* or *SCLT1* abolished the TF-localization of FBF1 as expected (Supplementary Figs. 9a and 10a). Remarkably, *CEP83* or *SCLT1* deficiency effectively inhibited IR-treated PML-NB translocation of FBF1, PML-NB upregulation (Supplementary Figs. 9a and 10b), and senescence responses (Supplementary Figs. 9b–e and 10c–e) in IMR-90 cells, strongly supporting TF localization of FBF1 is a prerequisite for senescence initiation.

## *Fbf1* ablation protects mice from IR-induced senescence and associated frailty

Cellular senescence and its associated SASP are causally implicated in age-related pathologies and cancer in humans[1,35]. It is thus critical to understand how stressed cells initiate senescence program and to use the knowledge to develop interventions to blunt deleterious senescence and improve healthspan/lifespan. Consistent results obtained from different mammalian cell types suggest an exciting and highly conserved paradigm that primary cilia, the sensory antennae of cells, use the ARL13B-ARL3-UBC9-FBF1 pathway to directly communicate with PML-NBs, the well-accepted nuclear signaling hub for stress responses, to initiate senescence program in stressed cells. To test the in-vivo importance of FBF1-regulated senescence, we subjected WT and *Fbf1[tm1a/tm1a]* mice to a sublethal dose of total-body IR treatment as

described[36]. Although *Fbf1* deficiency led to reduced survival of homozygous *Fbf1[tm1a/tm1a]* pubs likely due to the important role of cilia in embryonic development, the surviving *Fbf1[tm1a/tm1a]* mice were otherwise healthy without adverse phenotypes. *Fbf1[tm1a/tm1a]* mice develop progressive obesity due to enhanced adipogenesis, but are metabolically healthy, and even protected from premature death upon a high-fat diet treatment[26]. Like what we observed in cells, IR treatment greatly reduced the total SENP1 level, whereas upregulated PML in lung tissue of WT mice (Supplementary Fig. 11a). Of note, similar to what we observed in in vitro senescent cells (Supplementary Fig. 11b), IR-induced PML upregulation was abolished in *Fbf1* mice (Supplementary Fig. 11c). SUMOylation of endogenous FBF1 was also confirmed in lungs of IR-treated mice by immunoprecipitation (Supplementary Fig. 11d).

Strikingly, after IR treatment, WT but not *Fbf1[tm1a/tm1a]* mice showed accelerated aging phenotypes with massive hair greying and decreased body weight (Fig. 6a, b and Supplementary Fig. 11e). Immunohistochemistry also showed that p16[INK4a] and IL1α increased greatly in the liver of WT but not *Fbf1[tm1a/tm1a]* mice (Fig. 6c). Analysis of senescence and SASP markers revealed that senescence burden was much higher in the vital organs of WT, but not *Fbf1[tm1a/tm1a]* mice 2 weeks after IR treatment (Fig. 6d–g, and Supplementary Fig. 11f, g). To determine if *Fbf1* ablation protects animals from IR-associated long-term frailty, we measured muscle strength and physical endurance in animals 6 months after IR treatment. Of note, *Fbf1[tm1a/tm1a]* mice are healthy obese[26]. The increased body weight of *Fbf1[tm1a/tm1a]* mice may explain their reduced performance in RotaRod and treadmill even in non-IR-treated group (Fig. 6h, j). Nonetheless, only WT, but not *Fbf1[tm1a/tm1a]*, mice exhibited deteriorated endurance and frailty 6 months after IR treatment (Fig. 6h–j).

## Discussion

Here we reveal a paradigm that the suppression of a ciliary ARL13B-ARL3 GTPase cascade in stressed cells promotes the cilium-to-PML-NB translocation of FBF1, a decisive molecular event to drive senescence initiation (schematically shown in Fig. 5i). Genetic ablation of *Fbf1* effectively blocked IR-induced senescence responses and suppressed long-term health decline. Emerging evidence highlights the promising potential to target cellular senescence in age-related diseases and even cancer[5,37]. Our discoveries assign primary cilia a central role in senescence initiation and highlight primary cilia of stressed cells as promising targets to dampen deleterious senescence and improve healthspan/lifespan.

PML-NBs are highly dynamic and functional heterogeneity[8,38]. Although a small number of core proteins define the PML-NB (e.g., PML, DAXX, SP100, SUMO), hundreds of other proteins localize to PML-NBs in dynamic fashion, and certain PML-NB components undergo continuous exchange with the surrounding nucleoplasm[7,8]. Understanding the dynamic sequestration and function of PML-NB components, especially in stressed cells, is thus critical for understanding how PML-NBs carry out multifaceted stress-induced functions. The persistent colocalization of FBF1 and PML-NBs in stressed cells suggests that, like PML protein, FBF1 may shake hands with diverse interactors in different stages of senescence. At the ciliary base, FBF1 occupies the space between transition fibers[39], and forms a proteinaceous matrix-like barrier implicated in selective protein exchange[25,40,41]. It is tempting to speculate that FBF1 plays a conserved role to forms a matrix-like architecture on the surface of stress-induced PML-NBs and regulates the dynamic recruitment/exchange of diverse senescence effectors/regulators in stressed cells.

There have been contradictory observations that cilia either exist[42] or are absent[43] in senescent cells. Our discovery that primary cilia form transiently after stressor exposure highlights a strategy used in stressed cells by licensing ciliogenesis program to authorize the ARL-UBC9-FBF1 pathway after sensing irreparable damages. This combination likely allows irreversible senescence to be precisely controlled. Thus, primary cilia are not static but rather dynamic

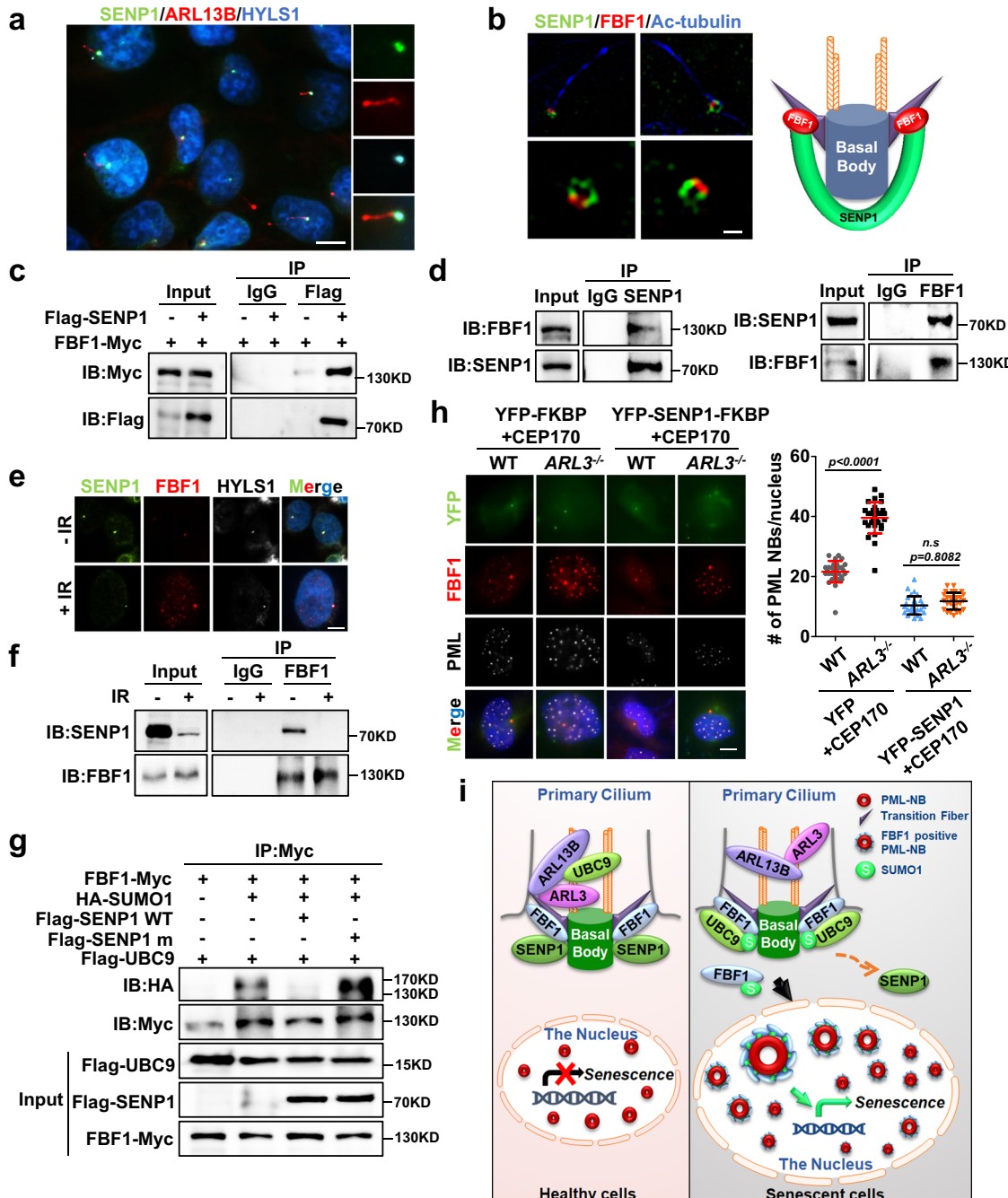

**Fig. 5 | DNA damage abolishes the ciliary SUMO protease SENP1 and enhances FBF1 SUMOylation. a** Immunofluorescent images for SENP1 staining in RCTE cells. Basal bodies were labeled with HYLS1. Scale bar, 10 μm. **b** Structured illumination microscopic images stained for FBF1 and SENP1 at ciliary base in IMR-90 cells. Scale bar, 1 μm. **c** Co-immunoprecipitation of SENP1 and FBF1 in 293T cells. **d** Endogenous SENP1 or FBF1 immunoprecipitates with FBF1 or SENP1 in RCTE cells. **e** Immunofluorescent images in RCTE cells at day 7 without or with irradiation treatment using antibodies against SENP1 and FBF1. Basal bodies were labeled with HYLS1. Scale bar, 10 μm. **f** Endogenous FBF1 immunoprecipitates with SENP1 in RCTE cells with or without irradiation. **g** SENP1 diminished FBF1 SUMOylation in a

co-immunoprecipitation assay in 293T cells. **h** Immunofluorescent images for FBF1, PML and YFP in WT or *ARL3⁻/⁻* RCTE cells co-transfected with CFP-FRB-CEP170c and YFP-FKBP-SENP1 plasmids at day 7 after irradiation. n=30 cells. Scale bar, 10 μm. **i** Proposed model: DNA-damage stress strongly downregulates ARLs, enables FBF1-UBC9 interaction, and results in FBF1 SUMOylation. SUMOylated FBF1 translocates from the ciliary base to PML-NBs, which is required for stress-induced PML-NB upregulation and senescence initiation. Three experiments were repeated independently with similar results (**a**–**h**). One-way ANOVA followed by Bonferroni multiple-comparison analysis was employed for **h**. Source data are provided as a Source Data file.

structures in stressed cells. It appears that the downregulation of the ciliary ARLs, from regulation of transcription/translation and/or protein stability, is enough to induce senescence by releasing UBC9 and promoting FBF1 SUMOylation, a process might be independent of sensory function of the primary cilium. However, whether and how

primary cilia of stressed cells can sense irreparable damage remains as intriguing questions. Cilia/flagella emerged in single-cell eukaryote organisms and evolved as sensory devices before the appearance of senescence programs in more sophisticated organisms. It remains plausible that sensory function of primary cilia could be hijacked by

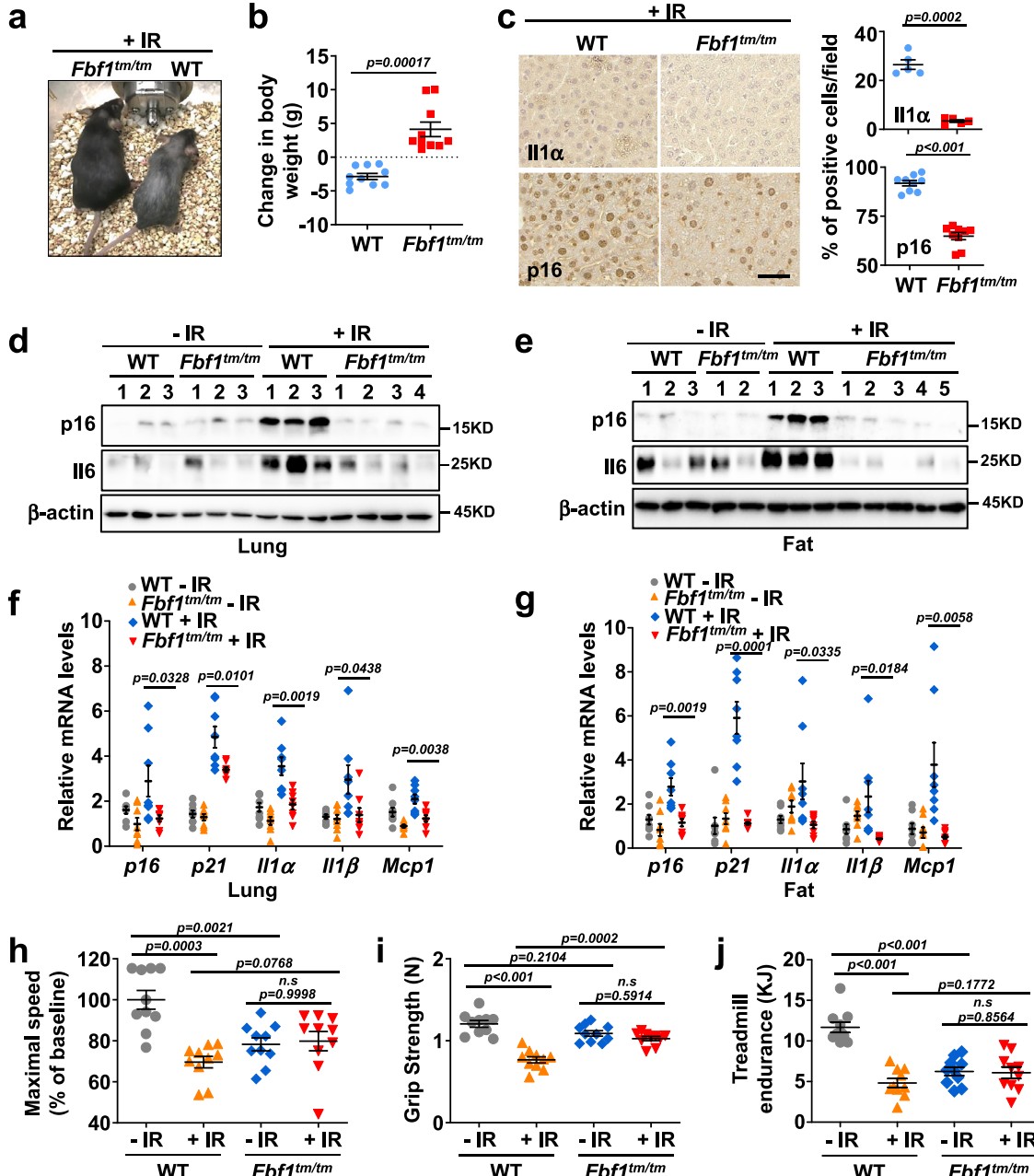

**Fig. 6 | _Fbf1_ ablation protects mice from IR-induced senescence and associated frailty. a** Representative images of mice 3 months after IR. **b** The change in body weight of mice with 3 months after IR. $n = 10$ mice. **c** at 2 weeks after IR, liver was analyzed by immunohistochemistry using Il1α antibody or p16 antibody and quantified; Scale bar: 200 μm. n >80 cells were counted in $n = 5$ different fields in $n = 3$ mice's Il1α staining and $n = 8$ different fields in $n = 4$ mice's p16 staining **d, e** Lysates obtained from lung (**d**) and fat (**e**) tissue of WT mice, or _Fbf1_$^{tm1a/tm1a}$ mice 2 weeks after IR were analyzed for the indicated proteins by western blot. **f, g** SASP-related genes were assessed by RT-qPCR in extracts of lung (**f**) and fat (**g**) tissue of

WT mice or _Fbf1_$^{tm1a/tm1a}$ mice 2 weeks after IR. $n = 8$ mice. Results from 3 independent experiments were statistically analyzed and plotted as means ± SEM. **h–j** Maximal speed (**h**), grip strength (**i**) and treadmill endurance (**j**) of 12-month-old WT mice or _Fbf1_$^{tm1a/tm1a}$ C57BL/6 mice, 6 months after exposure to mock or sublethal dose of total-body IR ($n = 10$ for both groups, 5 males and 5 females). Values are expressed as means ± SEMs. Two-tailed Student's unpaired _t_-test was used for analysis in **a**, **c**, **f** and **g**. One-way ANOVA followed by Bonferroni multiple-comparison analysis was employed for **h–j**. Source data are provided as a Source Data file.

stressed cells to sense stress cues, such as SASP components released autocrinally or paracrinally, to initiate senescence responses. Several studies indeed correlated cilia with SASP components in mammalian cells. For example, ciliogenesis of mesenchymal stromal cell is regulated by tumor necrosis factor-α[44]. IL-6 receptor IL-6Rα is restricted to primary cilia of epithelial cells[45]. To this end, determination of the repertoire and function of sensory receptors in the cilia of stressed cells holds great promise for deeper insights central to senescence regulation.

## Methods

### Animals

All animal experiments were performed according to protocols approved by the Institutional Animal Care and Use Committee (IACUC) at the Mayo Clinic (protocol A00053215–15). C57BL/6J mice were purchased from the Jackson Laboratory. The _Fbf1_$^{tm1a/tm1a}$ mice were obtained from Trust Sanger Institute and refreshed the genetic background as described[26]. The _Arl3_ floxed mouse line was a gift from Dr. Wolfgang Baehr's lab[46]. Mice were maintained in a pathogen-free

facility at 23–24 °C with relative humidity no higher than 50 percent under a 12 h light-dark cycle regimen with free access to a NCD (standard mouse diet Lab Diet 5053, St. Louis, MO) and water. Mice were bred and housed under a Specific Pathogen-Free (SPF) status. All mice were housed in static autoclaved HEPA-ventilated microisolator cages (27 × 16.5 × 15.5 cm) with autoclaved Enrich-o'Cobs (The Andersons Incorporated) for bedding. Cages and bedding were changed biweekly. Cages were opened only in class II biosafety cabinets. The majority of subjects from the study are the result of in-house mating, and littermate controls were used. Both male and female mice were used for the study, and the age was indicated in the figure legends.

Tissues from mice sacrificed at the indicated time points were snap-frozen in liquid nitrogen for biochemical studies or fixed in 4% paraformaldehyde for 24 h prior to processing and paraffin embedding. Paraffin-embedded tissues were cut into 5-μm sections.

## Cells and cell culture
IMR-90 (Cat# CCL-186), BJ cells (Cat# CRL-2522), RCTE (Cat# PCS-400-010) and HEK293T (Cat# CRL-3216) cells were purchased from ATCC. IMR-90 cells and BJ cells were cultured in EMEM (Eagle's minimum essential medium). RCTE cells were cultured in DMEM/F12. HEK293T cells were cultured in DMEM. The medium was supplemented with 10% fetal bovine serum and 1% penicillin and streptomycin. Cells were cultured under 5% $CO_2$ and 5% $O_2$ at 37 °C. MEFs were generated by standard methods as described previously[26]. Briefly, MEFs were generated from embryonic day 13.5 (E13.5) embryos of WT and mutant mice under the normal culture condition that includes DMEM with 10% FBS and 5% $CO_2$.

## Senescence and SA-β-Gal assays
Senescence was induced by irradiation. For ionizing radiation experiments, IMR-90 and BJ cells were exposed with 10 Gy by the X-ray irradiator (RS-2000 X-ray irradiator). RCTE or MEF cells were irradiated with 5 or 12 Gy X rays, respectively. The cells were harvested and analyzed 10 days later or otherwise indicated time. Control (proliferating) cells were mock irradiated. For $H_2O_2$ inducing senescence, sublethal amounts of $H_2O_2$ (200 μM) are added for 2 h at 37 °C. Prolonged $H_2O_2$ treatment is performed by adding 10 μM $H_2O_2$ with medium change after 3 days. Parallel cultured control cells grow in the media without $H_2O_2$. $H_2O_2$ is washed with PBS for terminating the treatment. Cells are kept on the incubation in normal medium for more 2 days. For IL1β inducing senescence, IMR-90 cells were treated with IL-1β (3 ng/ml) for 5 days. Senescent cells were identified by a senescence-associated β-galactosidase kit (Cell signaling) according to the manufacturer's protocol.

## Stable cell lines
Stable cell lines were constructed using lentivirus system. For producing lentiviral particles, pCDH or pLKO.1 construct, together with psPAX2 (Addgene #12260) and pMD2.G (Addgene #12259), was transfected into HEK293T cells using Fugene6 (Promega). Medium was replaced after 24 h, and virus was harvested 48 h post-transfection. Virus was filtered with a 0.45-μm OVDF filter (Millipore) and concentrated using Lenti-X Concentrator (Takara). Then, target cells were infected by lentivirus overnight with 10 μg/mL polybrene (Millipore), and medium was replaced. The cells were further selected by puromycin or G418 at 48 h after post-infection and fed with fresh puromycin or G418 every 2 days.

## Immunofluorescence microscopy
For FBF1 and PML-NB staining, cells were grown on glass coverslips and fixed with −20 °C methanol for 20 min, then blocked with 3% BSA and immune-stained with appropriate antibodies. For cilia staining, cells were fixed with 4% paraformaldehyde for 20 min at room temperature, followed by permeabilization with 0.1% Triton X-100 for 10 min. Cells were then blocked with 3% BSA and immune-stained with appropriate antibodies. Fluorescence images were acquired using Nikon TE2000-U with Metamorph software (Molecular Devices). ZEN - ZEISS Efficient Navigation, Nikon Elements (NIS ElementsAR ver. 4.6.0.) were used to acquire Immunofluorescence images.

## Antibodies
Primary antibodies: ARL13B (17711-1-AP; dilution 1:2000 for immunofluorescent and 1:1000 for western blotting), FBF1 (11531-1-AP; dilution 1:1000 for immunofluorescent and western blotting), ARL3 (10961-1-AP; dilution 1:1000 for immunofluorescent and western blotting), p16INK4a (10883-1-AP; dilution 1:1000 for western blotting), CEP83 (26013-1-AP; dilution 1:1000 for immunofluorescent and western blotting), SCLT1 (14875-1-AP; dilution 1:1000 for immunofluorescent and western blotting), IL6 (66146-1-Ig; dilution 1:1000 for western blotting), IL1α (16765-1-AP; dilution 1:200 for immunohistochemistry), KIF3A (13930-1-AP dilution 1:1000 for western blotting), IFT88 (13967-1-AP dilution 1:1000 for western blotting), His tag (66005-1-Ig; dilution 1:1000 for western blotting) from Proteintech; SUMO-1 (BML-PW8330-0025; dilution 1:1000 for western blotting) from Enzo Life Sciences. p16INK4a (ab54210; dilution 1:200 for immunohistochemistry) from Abcam; p16INK4a (SPC-775D; dilution 1:500 for western blotting) from StressMarq Biosciences; acetylated tubulin (T7451, dilution 1:5000 for immunofluorescent), β-actin (A1978; dilution 1:2000 for western blotting), FLAG tag (F1804; dilution 1:2000 for western blotting), HA tag (H3663; dilution 1:2000 for western blotting) and Myc tag (SAB2702192; dilution 1:2000 for western blotting) from Sigma; p21 (sc-6246; dilution 1:500 for western blotting), SUMO1 (D-11) (sc-5308; dilution 1:500 for western blotting), UBC9 (C-12) (sc-271057; dilution 1:500 for western blotting), PML (PG-M3) (sc-966; dilution 1:500 for immunofluorescent), PML (E-11) (sc-377390; dilution 1:200 for western blotting), HYLS1 (D-9) (sc-376721; dilution 1:200 for immunofluorescent), SENP1 (sc-271360; dilution 1:500 for immunofluorescent and western blotting), GST tag (B-14) (sc-138; dilution 1:1000 for western blotting) from Santa Cruz; ac-p53 (2525; dilution 1:1000 for western blotting) from Cell Signaling technology; glutamylated tubulin (GT335) (AG-20B-0020-C100; dilution 1:1000 for immunofluorescent) from AdipoGen Life Science.

Secondary antibodies: Peroxidase-AffiniPure Goat anti-mouse (111-035-144) or anti-rabbit (115-035-146) from Jackson ImmunoResearch Laboratories, dilution 1:2000 for western blotting. Goat anti-Rabbit IgG (H+L) Highly Cross-Adsorbed Secondary Antibody, Alexa Fluor 488 (A-11034) or 555 (A-21429), Goat anti-Mouse IgG1 Cross-Adsorbed Secondary Antibody, Alexa Fluor 555 (A-21127) or 488 (A-21121) or 647 (A-21240), Goat anti-Mouse IgG2a Cross-Adsorbed Secondary Antibody, Alexa Fluor 647 (A-21241), Goat anti-Mouse IgG2b Cross-Adsorbed Secondary Antibody, Alexa Fluor 647 (A-21242) from Invitrogen, dilution 1:1000 for immunofluorescent.

## DNA constructs and siRNAs
Human FBF1, SENP1, ARL3$^{WT}$, ARL3$^{DA}$, and ARL3$^{DN}$ were generated by PCR and subcloned into pCDH-Myc vector. CFP-FRB-CEP170c, YFP-FKBP12, and YFP-FKBP12-SENP1 were subcloned into pCDH vector. Human SUMO1, UBC9, SENP1, and SENP1 C603S were subcloned into pcDNA3-HA or pcDNA3-Flag vectors to generate HA-SUMO1, Flag-UBC9, Flag-SENP1 and Flag-SENP1 C603S constructs. Human FBF1 (598-1148 domain), ARL3 and UBC9 fragments were inserted into pET4T1 or pET28a vectors to generate GST-FBF1-CT, GST-ARL3, His-ARL3 or His-UBC9 plasmids, respectively. All constructs were verified by DNA sequencing. For construction of knockdown stable cell lines, shRNAs were inserted into pLKO.1-TRC plasmid, according to the Addgene instructions. siRNA duplexes were introduced into cells with Lipofectamine RNAiMAX (Invitrogen), following the manufacturer's manual. Typically, after siRNA transfection, cells were

collected for further analysis by western blot or immunofluorescence microscopy at the indicated time. RNAi Negative Control was purchased from GE Healthcare Dharmacon. siRNA oligonucleotides were obtained from Invitrogen. Sequences of shRNA or siRNA targeting corresponding mRNAs used are listed in Table S2 in Supplementary Data. 1. Sequences of *IFT88*, *CEP83*, *SCLT1* or *SENP1* siRNA were described in refs. [47–49].

## CRISPR-Cas9 gene editing

ARL3, ARL13B and FBF1 guide RNA (gRNA) was designed using an online tool (https://zlab.bio/guide-design-resources) and subcloned into pSpCas9(BB)−2A-GFP (px458), in which a green fluorescent protein (GFP) was fused to Cas9. RCTE cells were transfected with the gRNA construct for 48 h and subjected to flow cytometry to sort single GFP-positive cells. Single clones were expanded for further confirmation of the on-target cleavage using genomic PCR, followed by Sanger sequencing. gRNA sequences used are listed in Table S2 in Supplementary Data. 1.

## Immunoprecipitation assay and GST pull-down assay

Immunoprecipitation was performed using HEK293T or RCTE cell lysate in IP buffer (25 mM Tris-HCl, pH 7.4, 10 mM KCl, 1.5 mM MgCl$_2$, 1 mM EDTA, 1 mM EGTA, 150 mM NaCl, 0.5% NP-40), with complete protease inhibitor cocktail (Roche) added, following the manufacturer's manuals. The supernatant was collected by centrifugation for 20 min at 12,000 × $g$ at 4 °C and further pre-cleared using protein-G Sepharose for 4 h. After removal of protein-G beads, the pre-cleared supernatant was incubated with protein-G beads and 2 µg of the indicated primary antibodies or IgG control overnight at 4 °C. After washing, the Sepharose beads were boiled in 1× SDS-PAGE loading buffer. Proteins were detected by western blot. For GST pull-down assays, GST, the GST-fusion protein of FBF1 (C terminal) and 6× his-tagged UBC9 proteins were expressed in *Escherichia coli* BL21 strain and purified using glutathione or His resin Sepharose. Purified His-UBC9 proteins were incubated with GST or the GST-fusion proteins immobilized on glutathione Sepharose in binding buffer (25 mM Tris-HCl, pH 7.4, 150 mM NaCl, 1 mM EDTA, 10% glycerol, and 0.5% Triton X-100 and protease inhibitors) at 4 °C for 4 h. The beads were then washed with binding buffer and eluted in 1× SDS-PAGE loading buffer for further analysis by western blot.

## Western blot

Cells or tissues were homogenized in RIPA buffer with protease inhibitors (Roche). Protein concentration in each sample was measured using the BCA Pierce Protein assay kit (ThermoFisher) and normalized to the lowest concentration. Protein samples were subjected to standard SDS-PAGE gels and transferred to immuno-blot PVDF membranes (Millipore), then blocked with 5% nonfat dry milk and incubated overnight at 4 °C with primary antibodies. The secondary antibodies anti-mouse or anti-rabbit HRP-conjugated antibodies were incubated for 1 h at room temperature. After washing with TBS-T three times for 10 min each, horseradish peroxidase (HRP) substrates (ThermoFisher) were used to develop signals. Images were obtained using ChemiDoc™ Touch Imaging System (BIO-RAD). Images were analyzed using Image Lab (Bio-Rad) and ImageJ software.

## RNA isolation and qRT-PCR

For cultured cells, total RNA was extracted by TRIzol reagent (Invitrogene). cDNA was synthesized from the purified RNA using a high-capacity cDNA reverse transcription kit (Applied Biosystems). qPCR was performed using SYBR Green PCR Master Mix (BIO-RAD) with CFX384 real-time system (BIO-RAD). Tissue samples were lysed in Trizol reagent and homogenized for qRT-PCR. Real-Time PCR data were analyzed with CFX manager (Bio-Rad). PCR primers used are listed in Table S1 in Supplementary Data. 1.

## Immunohistochemistry

For immunohistochemistry staining, tissue sections (5 µm thickness) were deparaffinized in xylene and rehydrated through alcohol gradient, incubated with hydrogen peroxide (Dako) for 10 minutes to quench the endogenous peroxidase activity, and then rinsed with PBS three times. Antigen retrieval was performed using a steamer for 30 min in 0.1 M citrate buffer (pH 6.0). Serum-free protein block (Dako) was used to prevent nonspecific protein binding. Sections were then incubated with primary antibodies overnight at 4 °C, followed by a 60-min incubation with anti-rabbit secondary antibody at room temperature. The slides were developed with DAB Peroxidase Substrate Kit (Dako), counterstained with hematoxylin, dehydrated in ethanol, cleared in xylene, and then mounted with Permount mounting medium (Thermo Fisher Scientific).

## Cell viability assay

IMR-90 or MEF cells were plated as triplicates in 96-well plates and cultured for 48 h at 37 °C. After treatment with X-rays for indicated time, 10 µl of 12 mM 3-(4,5-dimethylthiazol-2-yl)−2,5-diphenyltetrazolium bromide (MTT) was added to each well and incubated for another 4 h. After replacing the culture medium with Stop Solution (40 mM HCL in isopropanol; 100 µl/well), the absorbance was measured at 590 nm on a plate reader. All-in-one microplate reader software Gen5 2.07 were used to acquire MTT assay results.

## TUNEL assay

Apoptosis of the cells in senescence was detected with a TUNEL assay kit (DeadEnd™ Fluorometric TUNEL System, G3250, Promega) following the manufacturer's protocol.

## In vitro SUMOylation

The in vitro SUMOylation kit reaction was purchased from Enzo Life Sciences. The reaction contains E1 and E2 enzymes, SUMO1/2/3, and GST-FBF1 (C terminal) proteins in SUMOylation buffer. SUMOylation reactions were incubated at 30 °C for 1 h. After termination with SDS-PAGE sample buffer, reaction products were subjected to SDS-PAGE.

## Chemically inducible cilia trapping system

The basis of the technology is chemically inducible dimerization, in which rapamycin induces the dimerization of FKBP-rapamycin binding domain (FRB) and FK506 binding protein (FKBP). To localize a fusion protein of YFP-FKBP-SENP1 to the ciliary base, we used the C terminus of CEP170. Transfected RCTE cells were incubated with 100 nM rapamycin for 30 min and then exposed with 5Gy X-rays for 5 days. Washout of rapamycin after 30 min did not affect protein dimerization in cells owing to the irreversible nature of the system.

## Total body irradiation

WT, *Fbf1^{tm1a/tm1a}* C57BL/6 mice of both sexes at 6 months of age were mock-irradiated or exposed to sub-lethal total body irradiation (4.25 Gy Gamma rays), and the indicated tissue was harvested at 14 days after the procedure for IHC, western blot or real-time PCR analysis. The physical function measurements were performed after 6 months post irradiation. Mice of both genders were randomly allocated to different groups and experiments were conducted non-blinded.

## Physical function measurements

Maximal walking speed was assessed using an accelerating RotaRod system. Mice were trained on the RotaRod for 2 days at speeds of 4, 6, and 8 rpm for 300 s. On the test day, mice were placed onto the RotaRod and then started at 4 rpm. The rotating speed was accelerated from 4 to 40 rpm over a 5-min interval. Testing consisted of three trials separated by a 30-min interval. Maximum trial length was 5 min, and results were averaged from three trials. Forelimb grip strength (N·F/kg)

was determined using a Grip Strength Meter. Results were averaged over four trials. For treadmill performance, mice were first acclimated to the treadmill for 3 consecutive days for 5 min each day starting at a speed of 5 m/min for 2 min, 7 m/min for 2 min, and then 9 m/min for 1 min at a 5% grade. On the test day, mice ran on the treadmill at an initial speed of 5 m/min for 2 min, and then the speed was increased by 2 m/min every 2 min until the mice were exhausted. Exhaustion will be defined as the inability of the mouse to remain on the treadmill despite an electrical shock stimulus and mechanical prodding. Running time was recorded and running distance and work (the product of body weight [kg], gravity [9.81 m/s2], vertical speed [m/s x angle], and time [s]) were calculated.

## Statistical analysis

Statistical significance was determined by unpaired Student's *t* test, one-way, or two-way ANOVA, accordingly. Statistical analysis was used with GraphPad Prism (version 5 and 9) and Microsoft Excel. The figure legends indicate the statistical test used.

## Reporting summary

Further information on research design is available in the Nature Portfolio Reporting Summary linked to this article.

## Data availability

No dataset was generated in this study. All data supporting the findings of this study are available in this article, its supplementary information, or the Source Data file. Source data are provided with this paper. All materials are available from the corresponding author upon request. Source data are provided with this paper.

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

## Acknowledgements

We thank Dr. Takanari Inoue (Stanford University) for sharing cilia trapping system and Dr. Wolfgang Baehr (U. Of Utah) for sharing *Arl3*$^{flox/flox}$ mice. This work was supported from the National Institutes of Health (NIH) research grants R01DK090038, R01DK099160, R01AG076469, P30 center grant P30DK90728, and the Mayo Clinic Robert M. and Billie Kelley Pirnie Translational Polycystic Kidney Disease Center and Mayo Clinic Foundation to J.H.; the Department of Defense (W81XWH2010214), and Pilot and Feasibility subawards from Mayo Clinic Translational PKD Center (P30DK90728) and Baltimore PKD Center (2P30DK090868) to K.L.; R01AG058812 to E.C.; R37 AG013925, P01 AG062413, R33 AG061456, the Connor Fund, Robert J. and Theresa W. Ryan, and the Noaber Foundation to J.K.

## Author contributions

Jing. H. generated the hypothesis and Jing. H., K.L., and Q.W. designed the experiments. X.M., Yi. Z., and Q.W. conducted most experiments. Q.W., Yi. Z., X.M., and Y.H. established and characterized the *Fbf1*$^{tm1a/tm1a}$ mice. X.M., Yi. Z., Y.H., Jie. H., X.Z., and N.L. contributed to the animal studies. X.M. and C.C. contributed to the immunofluorescence experiments. X.M. and D.Z. contributed to protein purification. X.M., Yu. Z., and K.H. contributed to the plasmid design and construction. K.L., Q.W., J.K., and E.C. contributed reagents and discussed data. X.M. and Jing. H. wrote the manuscript with input from all coauthors.

## Competing interests

The authors declare no competing interests.
