## [Peer Review File · Nature Communications]

A stress-induced cilium-to-PML-NB route drives senescence initiationREVIEWER COMMENTS

Reviewer #1 (Remarks to the Author):

Previous studies reported potential roles for ciliogenesis in the senescence response, yet a detailed understanding of how cilia are involved in senescence induction is still missing. In this study, Ma et al. report that ionizing radiation-induced senescence triggers robust, yet transient ciliogenesis in human lung fibroblast and mouse embryonic fibroblasts. The authors reveal the involvement of ARL13B-ARL3 GTPases negatively regulate SUMOylation of FBF1, and thereby suppress it from promoting PML nuclear body formation and consequently activating upregulation of various senescence factors and SA- β Gal. Lastly, the authors demonstrate that FBF1 knockout mice resist the development of senescent cells in response to ionizing radiation and consequently are resistant to IR-induced frailty. This is an interesting study with potentially important implications for suppressing the damaging effects of senescent cells on the organism. Despite this, a number of questions and concerns remain as outlined below.

Major concerns:

1. The authors convincingly demonstrate a role for cilia associated proteins (but not cilia) in IR-induced senescence in MEFs and human lung fibroblasts. Since the authors claim that cilia play a “central” role in senescence induction, possibly by sensing extracellular signals such as certain cytokines, they must show that other types of cellular senescence (replicative senescence, oncogene induced senescence, cytokine induced senescence, ROS, radiomimetics, ...) similarly require transient ciliogenesis for senescence induction. Otherwise, therapeutic opportunities would be limited to radiation induced disorders. In addition, more data needs to be shown that cilia formation, not merely certain proteins that are involved in cilia formation, is required for senescence induction. This is because it remains possible that ARL13B/ARL3/FBF1 play roles in senescence induction independent of their function in ciliogenesis.
2. Since human lung (Wi38, IMR90...) and skin (BJ, GM21...) fibroblasts activate different senescence programs (Rb vs p53), it would be important to test whether both cell types similarly require ciliogenesis to activate their senescence programs.
3. Some statement should be rephrased in order to prevent generalization of conclusions that may only apply to very specific conditions. These include any statements that ARL3, ciliogenesis, etc play a central or general role in senescence. Rather, these should be rephrased to state that this might be specific to IR-induced senescence (unless other evidence is provided). In addition, the statement that Fbf1 ablation suppressed long term healthspan decline is inaccurate because healthspan was not analyzed (discussion). That Cilia play a central role in senescence induction is not supported by the data since cilia were not analyzed, only certain proteins that also play a role in cilia formation/function, and other statements in the discussion.
4. Authors should be more specific when describing their results. Statements such as “impaired senescence induction”, “higher senescence responses”, “enhanced senescence responses” etc should be avoided, or supplemented with additional descriptions of the findings, since cellular senescence is a binary response: cells either undergo senescence or they do not. Enhanced/higher senescence

responses could imply that the same number of cells undergo senescence, yet expression of senescence markers are elevated. Alternatively it could mean that fewer cells undergo senescence, which is reflected in the lower abundance of senescence markers detected. These details should be provided.

5. More data should be shown to confirm that suppressing the formation of cilia also suppresses IR-induced senescence (or other types of senescence) induction. Since this is a central claim of the study, a WB demonstrating that p21/p16/p53 levels are reduced is not sufficient. Authors should demonstrate for example, that transient cilia formation is not detectable in IFT88 deficient cells in response to IR, or that other means to suppress cilia formation also impairs senescence induction.

6. Are ARL13B/ARL3/FBF1 deficient cells defective in ciliogenesis? These data should be shown.

7. Since cellular senescence is a critical tumor suppressing mechanism and FBF1 knockout impairs senescence, one would predict that FBF1 knockout mice develop more tumors compared to wt mice. Is this indeed the case? This should be discussed, especially in light of the suggestions by the authors that suppressing senescence by preventing cilia formation might improve healthspan and reduce frailty in old age.

Minor comments:

1. P5. Details are needed to evaluate what criteria were used to “screen dozens of ciliary proteins”. It would also be useful to name the dozens of proteins that were screened in a table and the results that were obtained.

2. Reference is needed that supports the claim that IFT88 is an essential protein for ciliogenesis.

3. Do IFT88 cells still undergo senescence? Is the kinetics of senescence induction impaired?

4. P6. It is unclear why the authors emphasize that FBF1 gradually translocates into the nucleus. Clearly all nuclear proteins have to translocate into the nucleus at one point, but this is probably not what the authors intended to convey. Is the novel finding that the translocation was gradual (if so, gradual compared to what?). Is this worth emphasizing since its translocation causes it to dissociate from cytoplasmic structures where consequently it impairs a cytoplasmic function? More details and data should be shown to clarify this.

Reviewer #2 (Remarks to the Author):

In this paper by Ma et al., the authors discover that irreparable stresses induce transient biogenesis of cilia, which are used by stressed cells to directly communicate with the promyelocytic leukemia nuclear bodies (PML-NBs) to initiate senescence responses. SUMOylated FBF1 translocates to PML-NBs to promote PML-NB-dependent senescence initiation and Fbf1 suppression effectively subdues global senescence burden and prevents associated health-span decline in irradiation-treated time. Overall, this is a well-written paper that provides novel and interesting data. While identifying key role of primary cilia in inducing senescence program, there are several concerns regarding this study that the authors

need to clarify. I feel that a substantial revision would be required prior to publication. Listed below are my specific comments.

(1) Why did the authors use irradiation (IR) among many stressors used in this field? What kind of stress signaling pathway is generated by IR? The authors fail to cite several other earlier studies about IR as a stressor.

(2) In Fig.1g-i, the authors conclude that ARL13B and ARL3 likely act in the same genetic pathway to regulate senescence. In Fig.1d and g, however, the pattern of β -gal staining looks different between ARL13B-knockdown cells and ARL3-knockdown cells. The authors cannot exclude the possibility that ARL13B and ARL3 might act in the different genetic pathway. The authors must be more careful in interpreting the data reported in Figure 1.

(3) FBF1 is a component of a transient fiber and therefore it is assumed that it is important for the formation of primary cilia. On page 10, the authors noted that Fbf1 knockout (KO) mice are healthy obese. However, the paper by Zhang et al demonstrated that approximately 50% of homozygous Fbf1 KO mice die before birth, and ~15% develop hydrocephalus, small size, and die within 4 weeks postnatally. It is assumed that the authors used the surviving Fbf1 KO mice, however, there are no descriptions in this paper. To avoid giving the impression that Fbf1 KO mice do not have any issues, I would suggest that the text reflect these issues. Most readers would consider Fbf1 total KO mice may not be suitable for the studies of muscle cells and therefore muscle cell-specific Cre mice should be introduced. More information could be provided on this, or at least some discussion of this issue provided. This point should at least be discussed in the case of experimental difficulties.

(4) A major shortcoming of this paper is that the authors fail to present convincing data that shows the involvement of PML-NBs in Fbf1 KO mice. The authors should observe the levels of SENP1, UBC9, and SUMOylation in Fbf1 KO mice. In addition, the number of PML-NBs in Fbf1 KO mice should be investigated.

Reviewer #3 (Remarks to the Author):

A stress-induced cilium-to-PML-NB route drives senescence initiation

Xiaoyu Ma, Yingyi Zhang, Yuanyuan Zhang, Xu Zhang, Yan Huang, Kai He, Chuan Chen, Jielu Hao¹, Debiao Zhao, Nathan K. LeBrasseur, James L. Kirkland, Eduardo N. Chini, Qing Wei, Kun Ling¹, Jinghua Hu

Reference NCOMMS-22-17861-T

Summary

This paper by Ma et al., proposes a role for the transient biogenesis of cilia to communicate with PML-NBs in the regulation of irradiation (IR) induced cellular senescence and senescence-associated secretory phenotype (SASP). While a role for PML-NBs in the regulation of cellular senescence is well established, the signalling mechanisms that influence PML-NB regulation of senescence remains to be fully defined. Here, the authors propose IR induced PML-NB senescence to be regulated by several proteins implicated in cilia formation/biogenesis via a mechanism that involves SUMOylation and the translocation of FBF1 to PML-NBs. The authors data implicate, although fail to prove in this reviewer's opinion, that cilia biogenesis directly contributes to stress response(s) induced by IR leading to senescence. For the authors conclusions to stand, they need to definitively demonstrate cilia biogenesis in response to IR; specifically as they state 'Our discoveries assign primary cilia a central role in senescence initiation and highlight primary cilia of stressed cells as promising targets to dampen deleterious senescence and improve healthspan/lifespan' [page 11]. The data presented is to a good standard, although some of the immunoprecipitation experiments and western blotting figures could be substantially improved to aid clarity and reader interpretation. The manuscript is generally well written, although some of the rationale used as a justification for experimentation is occasionally unclear or missing (at least for this reviewer) and its relevance lost to a general audience. While the data demonstrate a role for ARL138 and ARL3 in the negative regulation of FBF1 localization at PML-NBs that contributes to SASP, the data linking this to cilia biogenesis as a component of an IR induced stress response to regulate cellular senescence is indirect and noticeably absent considering the title of the manuscript (see above). Substantial revisions would be required in order to experimentally demonstrate the authors major conclusions as the manuscript currently stands.

Major comments:

1. In this reviewer's opinion, the authors fail to show that IR leads to cilia biogenesis in the manuscripts predominant model system (primary lung fibroblasts; IMR-90 cells). The data as presented, quantitation of Ac-Tubulin by indirect immunofluorescence (fig 1a), is insufficient to support the notion that IR induces cilia formation in these cells relative to other cellular structures that contain tubulin. Notably, the authors present data to suggest that $\geq 20\%$ of IMR-90 cells are ciliated at day 0 post-IR treatment (Fig 1a). Importantly, many of the proteins the authors claim to be associated with cilia biogenesis are constitutively expressed in a range of cell types independently of IR (e.g., <https://www.proteinatlas.org/ENSG00000188878-FBF1/cell+line>). Thus, the authors should demonstrate by EM that the structures measured by immunofluorescence are specifically cilia for their main conclusions to stand. Notably, the authors frequently interchange between cell types (IMR-90 [fibroblast] vs RCTE [epithelial] cells); making it difficult to independently establish whether the phenotypes described are specific to the differentiation status of the cell. Where alternate cell types are used, the authors should make a clear rationale argument for their incorporation into the study.
2. Data relating to the role of FBF1 post-translational modification by SUMOylation is lacking. Many of the assays only demonstrate an interaction with SUMO and/or Ubc9 in the presence or absence of ARL3. No convincing data is present to show FBF1 is directly SUMOylated, the percentage of endogenous protein to undergo SUMOylation that results in its localization at PML-NBs, nor FBF1 residues that become modified by SUMO. Thus, it is equally plausible that FBF1 is recruited to PML-NBs via non-

covalent SUMO-SIM interactions. While it's clear that a role for SUMOylation is likely to mediate the localization of FBF1 at PML-NBs upon IR, the precise mechanism remains unclear and is currently ambiguous (at best) or misleading (at worse).

Minor comments:

1. The authors need to present evidence to justify the statement 'We discovered that irradiation (IR) induces robust but transient ciliogenesis in stressed cells' [page 3]. In this reviewer's opinion, this statement (and similar analogous statements throughout the manuscript) is not adequately supported by the evidence provided. Depletion expms of IFT88 (as presented) are not sufficient to confirm the presence/absence of ciliogenesis. Such inadequacies to address ciliogenesis are amplified by the statement 'There have been contradictory observations that cilia either exist or are absent in senescent cells' [page 11]. Thus, the authors need to definitively prove that ciliogenesis occurs in their model system for their major conclusions to stand.
2. The number of independent biological replicates for each experiment presented in the manuscript is missing, making independent interpretation of the data difficult to establish. Figure legends should contain this information and the corresponding statistical analysis performed for clear interpretation.
3. Many of the WBs are unreasonably cropped for studies investigating FBF1 SUMOylation and lack appropriate molecular mass markers, making independent interpretation as to the MW shift associated with protein SUMOylation impossible to determine. Evidence should be provided that FBF1 undergoes covalent SUMOylation and the proportion of molecules SUMOylated at an endogenous level to promote re-localization to PML-NBs.
4. A stronger rationale needs to be presented for the information described for CEP164 (page 6). It is currently unclear why this protein is even mentioned in this study. Revision of the manuscript is strongly recommended.
5. Data should be provided to support the statement 'Strikingly, ablation of FBF1 completely suppressed PML-NB upregulation in IR-treated IMR-90 cells' [page 7]. No evidence is present for the upregulation of PML or PML-NB constituent proteins, only for the number of bodies to change in their relative frequency per nuclei.
6. Data should be provided to support the statement 'In agreement with the critical role of cilia in senescence regulation' [page 7]. No data is presented to establish a critical role for cilia, only IFT88 depletion. It remains plausible that the mechanism described may be IFT88-dependent but independent of cilia or cilia biogenesis.
7. Evidence should be provided to support the statement 'TF localization is a prerequisite for PML-NB translation of FBF1 in stressed cells' [page 9]. Currently there is no evidence to suggest a role in translation, only translocation of FBF1 to PML-NBs.

REVIEWER COMMENTS

Reviewer #1 (Remarks to the Author):

Previous studies reported potential roles for ciliogenesis in the senescence response, yet a detailed understanding of how cilia are involved in senescence induction is still missing. In this study, Ma et al. report that ionizing radiation-induced senescence triggers robust, yet transient ciliogenesis in human lung fibroblast and mouse embryonic fibroblasts. The authors reveal the involvement of ARL13B-ARL3 GTPases negatively regulate SUMOylation of FBF1, and thereby suppress it from promoting PML nuclear body formation and consequently activating upregulation of various senescence factors and SA-bGal. Lastly, the authors demonstrate that FBF1 knockout mice resist the development of senescent cells in response to ionizing radiation and consequently are resistant to IR-induced frailty. This is an interesting study with potentially important implications for suppressing the damaging effects of senescent cells on the organism. Despite this, a number of questions and concerns remain as outlined below.

Major Comments:

1. The authors convincingly demonstrate a role for cilia associated proteins (but not cilia) in IR-induced senescence in MEFs and human lung fibroblasts. Since the authors claim that cilia play a “central” role in senescence induction, possibly by sensing extracellular signals such as certain cytokines, they must show that other types of cellular senescence (replicative senescence, oncogene induced senescence, cytokine induced senescence, ROS, radiomimetics, ...) similarly require transient ciliogenesis for senescence induction. Otherwise, therapeutic opportunities would be limited to radiation induced disorders. In addition, more data needs to be shown that cilia formation, not merely certain proteins that are involved in cilia formation, is required for senescence induction. This is because it remains possible that ARL13B/ARL3/FBF1 play roles in senescence induction independent of their function in ciliogenesis.

We completely agree with the reviewer for the insightful critiques. To address whether the primary cilium is required for other types of cellular senescence, we examined the role of the primary cilium in oxidative stressor (H₂O₂) or inflammatory stressor (IL1 β) induced senescence. As shown in the new **Supplementary Fig. 1**, like IR-treatment, transient ciliogenesis was detected in either oxidative stressor (H₂O₂) or inflammatory stressor (IL1 β) induced senescence in IMR90 cells. FBF1 deficiency can also effectively blocked H₂O₂- or IL1β- induced senescence responses (new **Supplementary Fig. 3G-I and 4C-D**). Collectively, our new data demonstrated that primary cilia indeed play a central role in senescence induced by different stressors.

To address if cilia formation is required for senescence induction, we knocked down *KIF3A* and *IFT88*, two essential structural components for cilia formation. *siKIF3A* and *siIFT88* treatment consistently suppress both ciliogenesis and senescence responses in IMR-90 cells exposed to IR, H₂O₂, or IL1β treatment (new **Supplementary Fig. 1**). Consistently, cilia ablation by *siKIF3A* or *siIFT88* suppressed PML-NB translocation of FBF1 and PML-NB upregulation in either IR, H₂O₂-, or IL1β-treated cells (new **Supplementary Fig. 54F-H**). Further, ARL13B-deficiency

significantly upregulates PML-NBs and senescence responses in IR-treated cells, which can be abolished by cilia suppression (new **Supplementary Fig. 5**).

2. Since human lung (Wi38, IMR90...) and skin (BJ, GM21...) fibroblasts activate different senescence programs (Rb vs p53), it would be important to test whether both cell types similarly require ciliogenesis to activate their senescence programs.

To examine if cilia-regulated senescence is also true in other cell types, we tested skin fibroblast BJ cells as the reviewer suggested. As shown in the new **Supplementary Fig. 6**, ARL3/ARL13B/FFB1 play same roles in senescence induction in BJ cells. Specifically, deficiency of *ARL3* or *ARL13B* strongly promoted PML-NB translocation of FFB1, PML-NB biogenesis and senescence responses, whereas *FFB1* deficiency suppressed PML-NB biogenesis and senescence responses in stressed BJ cells (new **Supplementary Fig. 6**).

3. Some statement should be rephrased in order to prevent generalization of conclusions that may only apply to very specific conditions. These include any statements that ARL3, ciliogenesis, etc play a central or general role in senescence. Rather, these should be rephrased to state that this might be specific to IR-induced senescence (unless other evidence is provided). In addition, the statement that Fbfl ablation suppressed long term healthspan decline is inaccurate because healthspan was not analyzed (discussion). That Cilia play a central role in senescence induction is not supported by the data since cilia were not analyzed, only certain proteins that also play a role in cilia formation/function, and other statements in the discussion.

We heartily appreciate the reviewer critiques. With the aforementioned new data achieved by the experiments suggested by the reviewer, our conclusion is greatly strengthened in revised manuscript. For cilia analysis, we included Glutamylated-tubulin as a specific cilia marker and SCLT1 as a marker to label the ciliary base (new **Supplementary Fig. 1**). We also carefully revised the manuscript to avoid overstatement as suggested.

4. Authors should be more specific when describing their results. Statements such as “impaired senescence induction”, “higher senescence responses”, “enhanced senescence responses” etc should be avoided, or supplemented with additional descriptions of the findings, since cellular senescence is a binary response: cells either undergo senescence or they do not. Enhanced/higher senescence responses could imply that the same number of cells undergo senescence, yet expression of senescence markers are elevated. Alternatively it could mean that fewer cells undergo senescence, which is reflected in the lower abundance of senescence markers detected. These details should be provided.

We thank the reviewer for the constructive comment. We revised the manuscript carefully to focus on the change of senescence markers and avoid statements against the view that senescence is a binary process.

5. More data should be shown to confirm that suppressing the formation of cilia also suppresses IR-induced senescence (or other types of senescence) induction. Since this is a central claim of the study, a WB demonstrating that p21/p16/p53 levels are reduced is not sufficient. Authors should demonstrate for example, that transient cilia formation is not detectable in IFT88 deficient cells

in response to IR, or that other means to suppress cilia formation also impairs senescence induction.

We agree with the reviewer that this is an important question. As aforementioned, we performed new experiments by knocking down essential cilia structural component *KIF3A* and *IFT88* and tested the impacts in IR-, H₂O₂-, or IL1 β -induced senescence. Our newly obtained data confirm cilia loss in these treatments and support that cilia suppression impairs senescence induced by different stressors (new **Supplementary Fig. 1, 4, and 5**). Moreover, ARL13B-deficiency significantly upregulates PML-NBs and senescence responses in IR-treated cells, which can be abolished by cilia suppression (new **Supplementary Fig. 5**).

6. Are ARL13B/ARL3/FBF1 deficient cells defective in ciliogenesis? These data should be shown.

We are sorry we did not make it clear. The impact of ARL13B/ARL3/FBF1 deficiency on mammalian cilia formation have been previously reported by our and other labs. Specifically, ARL13B and FBF1 deficiency led to truncated cilia, whereas ARL3 deficiency does not affect ciliogenesis. The impacts on IMR-90 cells are same as on other cell types (**Fig. R1**).

7. Since cellular senescence is a critical tumor suppressing mechanism and FBF1 knockout impairs senescence, one would predict that FBF1 knockout mice develop more tumors compared to wt mice. Is this indeed the case? This should be discussed, especially in light of the suggestions by the authors that suppressing senescence by preventing cilia formation might improve healthspan and reduce frailty in old age.

This is a great question. Cellular senescence and associated SASP are thought to be a double-edge sword highly dependent on context and cell type and variable during the different stages of cancer progression in tumorigenesis, could be either pro-oncogenic or anti-oncogenic. Recently, senolytic drugs have been widely explored in treating cancers in rodent models. Our preliminary analyses showed that *Fbfl* mice possess lower senescence, lower inflammation, improved lifespan but not increased tumorigenesis, even under high-fat-diet treatment. This is in consistent with previous studies that genetic or pharmacological clearance of senescent cells effectively improves survival and healthspan in rodent models. Of note, emerging evidence suggest that primary cilia are also implicated in tumorigenesis and context based. Cilia ablation can either inhibit or promote tumorigenesis in different models. Thus, it is interesting to investigate how cilia-regulated tumorigenic pathways and cilia-regulated senescence functionally crosstalk *in vivo*. It is

Fig. R1. Immunofluorescent images showing primary cilia in ARL13B-, ARL3- or FBF1-knockdown IMR-90 cells. Anti-Glu-tubulin and anti-ARL13B was used to label cilia proper, HYLS1 was used to label the ciliary base. Scale bar, 10 μ m.

conceivable that the underlying regulation could be complicated at multiple levels and probably depend on cell type and/or cancer-context.

Minor comments:

1. P5. Details are needed to evaluate what criteria were used to “screen dozens of ciliary proteins”. It would also be useful to name the dozens of proteins that were screened in a table and the results that were obtained.

We revised the manuscript to specify the proteins we screened (**Page 5** in new manuscript).

2. Reference is needed that supports the claim that IFT88 is an essential protein for ciliogenesis.

We are sorry for the missing references and included it in revised version.

3. Do IFT88 cells still undergo senescence? Is the kinetics of senescence induction impaired?

Yes, *IFT88* deficiency impairs both ciliogenesis and senescence in response to different stressors (new **Supplementary Fig. 1, 4**).

4. P6. It is unclear why the authors emphasize that FBF1 gradually translocates into the nucleus. Clearly all nuclear proteins have to translocate into the nucleus at one point, but this is probably not what the authors intended to convey. Is the novel finding that the translocation was gradual (if so, gradual compared to what?). Is this worth emphasizing since its translocation causes it to dissociate from cytoplasmic structures where consequently it impairs a cytoplasmic function? More details and data should be shown to clarify this.

We are sorry for the confusion. We observed that FBF1 protein level and the nuclear translocation of FBF1 increases continually in a few days during senescence progression. This observation led us to convey the observation that this translocation happens in several days but not in an acute way. We revised the manuscript to avoid the confusion (**Page 7** in new manuscript).

Reviewer #2 (Remarks to the Author):

In this paper by Ma et al., the authors discover that irreparable stresses induce transient biogenesis of cilia, which are used by stressed cells to directly communicate with the promyelocytic leukemia nuclear bodies (PML-NBs) to initiate senescence responses. SUMOylated FBF1 translocates to PML-NBs to promote PML-NB-dependent senescence initiation and Fbfl suppression effectively subdues global senescence burden and prevents associated health-span decline in irradiation-treated time. Overall, this is a well-written paper that provides novel and interesting data. While identifying key role of primary cilia in inducing senescence program, there are several concerns regarding this study that the authors need to clarify. I feel that a substantial revision would be required prior to publication. Listed below are my specific comments.

1. Why did the authors use irradiation (IR) among many stressors used in this field? What kind of stress signaling pathway is generated by IR? The authors fail to cite several other earlier studies about IR as a stressor.

We are sorry for not clarifying the rationale to use IR treatment to induce senescence. We revised the manuscript to include relevant references. We further rigorously tested our central hypothesis in ROS and cytokine-induced senescence, and our new data demonstrate that primary cilia and ARL3/ARL13B/FBF1 are generally required for different senescence types. The new data are included as supplementary material in revised manuscript (new **Supplementary Fig. 1, 3, 4, 5, 6**). Regarding to what kind of cues are sensed by cilia to initiate the senescence, our hypothesis is that SASP factors may promote senescence autocrinally. We are actively investigating this now.

2. In Fig. 1g-i, the authors conclude that ARL13B and ARL3 likely act in the same genetic pathway to regulate senescence. In Fig. 1d and g, however, the pattern of β -gal staining looks different between ARL13B-knockdown cells and ARL3-knockdown cells. The authors cannot exclude the possibility that ARL13B and ARL3 might act in the different genetic pathway. The authors must be more careful in interpreting the data reported in Figure 1.

We are sorry for the confusion. We repeated the SA- β -gal staining three times. All the results showed that ARL13B-knockdown cells and ARL3-knockdown cells increased the percentage of SA- β -gal staining positive cells compared to control cells. The cell density of ARL3-knockdown and ARL13B-knockdown experiments is different, and this might be the reason why they look different. We hypothesized that ARL13B and ARL3 may act in the same genetic way because of the existing evidence that ARL3 and ARL13B GTPases mutated in the same genetic disease Joubert syndrome, the observation that they were both downregulated in senescence, and our previous studies that ARL13B activates ARL3 as a GTPase activating protein. To confirm if ARL13B and ARL3 act in the same pathway, we later demonstrated the requirement of the ARL13B-ARL3 GTPase cascade in FBF1-mediated senescence induction. To avoid confusion, we revised the manuscript to soften the statement.

3. FBF1 is a component of a transient fiber and therefore it is assumed that it is important for the formation of primary cilia. On page 10, the authors noted that Fbfl knockout (KO) mice are healthy obese. However, the paper by Zhang et al demonstrated that approximately 50% of

homozygous Fbfl KO mice die before birth, and ~15% develop hydrocephalus, small size, and die within 4 weeks postnatally. It is assumed that the authors used the surviving Fbfl KO mice, however, there are no descriptions in this paper. To avoid giving the impression that Fbfl KO mice do not have any issues, I would suggest that the text reflect these issues. Most readers would consider Fbfl total KO mice may not be suitable for the studies of muscle cells and therefore muscle cell-specific Cre mice should be introduced. More information could be provided on this, or at least some discussion of this issue provided. This point should at least be discussed in the case of experimental difficulties.

We are sincerely sorry for not making this clear. We revised the manuscript accordingly to clarify the phenotypes of *Fbfl* KO mice as suggested by the reviewer. Overall, *Fbfl* deficiency appears to be detrimental for embryonic development but dispensable to viability and health once the animals survive. Our unpublished data show that all survived *Fbfl* mice live even a longer life (~20% increased longevity) when compared with WT littermates. We have other independent projects also analyzed the muscle, fat, bone, and kidney cells and see no detrimental effects. We believe this longevity is at least partly caused by systematically low senescence. We want to ask a favor to not put all our preliminary data mentioned here in this manuscript as we are preparing a separate study regarding the role of cilia-mediated senescence in longevity. We did revise the manuscript to give more detailed description about this mouse model (**Page 10** in new manuscript).

(4) A major shortcoming of this paper is that the authors fail to present convincing data that shows the involvement of PML-NBs in Fbfl KO mice. The authors should observe the levels of SENP1, UBC9, and SUMOylation in Fbfl KO mice. In addition, the number of PML-NBs in Fbfl KO mice should be investigated.

We completely agree with the reviewer. In past years, we have tried numerous PML antibodies to stain *in vivo* PML-NBs in mouse tissues, unfortunately, we face a technical challenge that there is no good antibody available to detect PML-NBs even in cultured mouse cells in immunofluorescence assays. The background noise and low sensitivity made us couldn't distinguish the PML-NBs clearly. After investing significant efforts, we finally optimized immunoblotting protocol by using PML antibody in western blotting for mouse proteins. Like what we observed in human cells, IR-treatment greatly reduced the total SENP1 level, whereas upregulated PML in lung tissue (**Supplementary Fig. 11A**). Of note, IR-induced PML upregulation was abolished in *Fbfl* mice (**Supplementary Fig. 11C**). Furthermore, change of endogenous FBF1 SUMOylation was also confirmed in lungs of IR-treated mice by immunoprecipitation (**Supplementary Fig. 11D**).

Reviewer #3 (Remarks to the Author):

A stress-induced cilium-to-PML-NB route drives senescence initiation

Xiaoyu Ma, Yingyi Zhang, Yuanyuan Zhang, Xu Zhang, Yan Huang, Kai He, Chuan Chen, Jieli Hao1, Debiao Zhao, Nathan K. LeBrasseur, James L. Kirkland, Eduardo N. Chini, Qing Wei, Kun Ling1, Jinghua Hu

Reference NCOMMS-22-17861-T

Summary

This paper by Ma et al., proposes a role for the transient biogenesis of cilia to communicate with PML-NBs in the regulation of irradiation (IR) induced cellular senescence and senescence-associated secretory phenotype (SASP). While a role for PML-NBs in the regulation of cellular senescence is well established, the signalling mechanisms that influence PML-NB regulation of senescence remains to be fully defined. Here, the authors propose IR induced PML-NB senescence to be regulated by several proteins implicated in cilia formation/biogenesis via a mechanism that involves SUMOylation and the translocation of FBF1 to PML-NBs. The authors data implicate, although fail to prove in this reviewer's opinion, that cilia biogenesis directly contributes to stress response(s) induced by IR leading to senescence. For the authors conclusions to stand, they need to definitively demonstrate cilia biogenesis in response to IR; specifically as they state 'Our discoveries assign primary cilia a central role in senescence initiation and highlight primary cilia of stressed cells as promising targets to dampen deleterious senescence and improve healthspan/lifespan' [page 11]. The data presented is to a good standard, although some of the immunoprecipitation experiments and western blotting figures could be substantially improved to aid clarity and reader interpretation. The manuscript is generally well written, although some of the rationale used as a justification for experimentation is occasionally unclear or missing (at least for this reviewer) and its relevance lost to a general audience. While the data demonstrate a role for ARL138 and ARL3 in the negative regulation of FBF1 localization at PML-NBs that contributes to SUSP, the data linking this to cilia biogenesis as a component of an IR induced stress response to regulate cellular senescence is indirect and noticeably absent considering the title of the manuscript (see above). Substantial revisions would be required to in order to experimentally demonstrate the authors major conclusions as the manuscript currently stands.

Major comments:

1. In this reviewer's opinion, the authors fail to show that IR leads to cilia biogenesis in the manuscripts predominant model system (primary lung fibroblasts; IMR-90 cells). The data as presented, quantitation of Ac-Tubulin by indirect immunofluorescence (fig 1a), is insufficient to support the notion that IR induces cilia formation in these cells relative to other cellular structures that contain tubulin. Notably, the authors present data to suggest that $\geq 20\%$ of IMR-90 cells are ciliated at day 0 post-IR treatment (Fig 1a). Importantly, many of the proteins the authors claim to be associated with cilia biogenesis are constitutively expressed in a range of cell types independently of IR (e.g., <https://www.proteinatlas.org/ENSG00000188878-FBF1/cell+line>). Thus, the authors should demonstrate by EM that the structures measured by immunofluorescence are specifically cilia for their main conclusions to stand. Notably, the authors frequently

interchange between cell types (IMR-90 [fibroblast] vs RCTE [epithelial] cells); making it difficult to independently establish whether the phenotypes described are specific to the differentiation status of the cell. Where alternate cell types are used, the authors should make a clear rationale argument for their incorporation into the study.

We thank the reviewer for the insightful critique. In the original submission, we used anti-acetylated Tubulin to label cilia because it can be used together with other antibodies used in **Fig. 1**. We indeed used EM to confirm IR-induced ciliogenesis in IMR-90 cells, but the limited view field for EM prevents us to quantitatively analyze ciliogenesis during senescence progression. To better address the reviewer's concern, we performed additional experiments by using anti-Glutamylated tubulin as a specific axoneme marker, and HYLS1 as a specific basal body marker, to co-stain with Ac-tubulin in IR-treated IMR90 cells. As shown in the **Fig. R2**, we confirmed transient ciliogenesis during senescence induction, which is also consistent with the observation made in original **Fig. 1A**. Further, we included new data in the new **Supplementary Fig. 1**, like IR-treatment, transient ciliogenesis was detected in either oxidative stressor (H_2O_2) or inflammatory stressor ($IL1\beta$) induced senescence in IMR90 cells.

Fig. R2. Immunofluorescent images for changes of primary cilia in IMR-90 cells after senescence-induction by 10 Gy X-ray's irradiation. Scale bar, 10 μ m.

Regarding the different cell types used in this manuscript, we sincerely apologize for the confusion. For senescence cell models, we chose widely used human fetal lung fibroblasts (IMR-90), mouse MEFs, and human renal collecting tubule epithelia (RCTE) cells. We want to understand if the novel phenotype we observed are conserved in different species and different cell types. The robustness of RCTE cells enables challenging experiments such as transfection, biochemical analyses, genome editing, or isolations of cell clones, which are otherwise notoriously challenging in primary fibroblast IMR-90 cells or MEFs due to their limited population doublings. Further, our new data support that the function of the ARL-FBF1-PML-NBs pathway in senescence regulation is highly conserved not only in IMR-90, MEFs, and RCTE cells, but also in the new skin fibroblasts BJ cells (suggested by the other reviewer) including in the revised manuscript (the new **Supplementary Fig. 6**). In unrelated projects, we also confirm the conservation of this pathway in senescence regulation in mouse retinal epithelial cells and mouse preadipocytes and bone progenitors. We are confident to conclude that the essential role of primary cilia in senescence regulation is a general mechanism but not cell type dependent in both mice and human. To avoid the misunderstanding, we clarify the rationale to use different cell types in revised manuscript and describe cell types in methods and figure legend sections in revised manuscript.

2. Data relating to the role of FBF1 post-translational modification by SUMOylation is lacking. Many of the assays only demonstrate an interaction with SUMO and/or Ubc9 in the presence or absence of ARL3. No convincing data is present to show FBF1 is directly SUMOylated, the percentage of endogenous protein to undergo SUMOylation that results in its localization at PML-NBs, nor FBF1 residues that become modified by SUMO. Thus, it is equally plausible that FBF1 is recruited to PML-NBs via non-covalent SUMO-SIM interactions. While it's clear that a role for SUMOylation is likely to mediate the localization of FBF1 at PML-NBs upon IR, the precise mechanism remains unclear and is currently ambiguous (at best) or misleading (at worse).

We completely agree with the reviewer, and this is one of the most important questions we are pursuing in our current research plan. Our evidence demonstrate that TF localization is a prerequisite for PML-NB translocation of FBF1 in stressed cells, and a cilia-base-trapped SUMO hydrolase SENP1 effectively block both PML-NB translocation of FBF1 and PML-NB upregulation in senescent cells. In revised manuscript, we performed experiments to confirm FBF1 is indeed SUMOylated by *in-vitro* SUMOylation assay (**Fig. 4A**). We further used immunoprecipitation (IP) assay to confirm endogenous FBF1 SUMOylation, which increased greatly upon IR treatment (**Fig. 4C**). In an independent project to map down FBF1 SUMOylation site(s), we have generated numerous FBF1 mutants by site-directed mutagenesis. Our preliminary data (**Fig. R3 and data not shown**) have identified Lysine 975 of FBF1 is one of the FBF1 SUMOylation site. We ask for a favor to make this discovery confidential because we are performing more experiments to understand its *in vivo* physiological importance and have identified potential but intriguing players that recognize SUMOylated FBF1 to enable its nuclear translocation. We do agree with the reviewer that we should be more careful to propose the role of SUMOylation in FBF1 translocation to avoid the misleading. We thus revised the manuscript accordingly to avoid misleading (**Page 8** in new manuscript).

Fig. R3. FBF1 is SUMOylated at K975. **A.** K975R mutant diminished FBF1 SUMOylation in a co-immunoprecipitation assay in 293T cells. **B.** *In-vitro* SUMOylation assay of FBF1 and western blotting using a SUMO1 antibody. Myc-tagged FBF1 was purified from 293T cells.

Minor comments:

1. The authors need to present evidence to justify the statement ‘We discovered that irradiation (IR) induces robust but transient ciliogenesis in stressed cells’ [page 3]. In this reviewer’s opinion, this statement (and similar analogous statements throughout the manuscript) is not adequately supported by the evidence provided. Depletion expms of IFT88 (as presented) are not sufficient to confirm the presence/absence of ciliogenesis. Such inadequacies to address ciliogenesis are amplified by the statement ‘There have been contradictory observations that cilia either exist or

are absent in senescent cells' [page 11]. Thus, the authors need to definitively prove that ciliogenesis occurs in their model system for their major conclusions to stand.

We are sorry for inadequate description that cause the confusion. As shown in the new **Supplementary Fig. 1A and B**, by using specific cilia markers, like IR-treatment, transient ciliogenesis was detected in either oxidative stressor (H₂O₂) or inflammatory stressor (IL1 β) induced senescence in IMR90 cells. To further address if cilia formation is required for senescence induction, we knocked down *KIF3A* and *IFT88*, two essential structural components for cilia formation. *siKIF3A* and *siIFT88* treatment consistently suppress both ciliogenesis and senescence responses in IMR-90 cells exposed to IR, H₂O₂, or IL1 β (new **Supplementary Fig. 1C-I**). Consistently, Cilia ablation by *siKIF3A* or *siIFT88* suppressed PML-NB translocation of FBF1 and PML-NB upregulation in either IR, H₂O₂-, or IL1 β - treated cells (new **Supplementary Fig. 4F-H**). Further, ARL13B-deficiency significantly upregulates PML-NBs and senescence responses in IR treated cells, which can be abolished by cilia suppression (new **Supplementary Fig. 5**).

2. The number of independent biological replicates for each experiment presented in the manuscript is missing, making independent interpretation of the data difficult to establish. Figure legends should contain this information and the corresponding statistical analysis performed for clear interpretation.

We are sincerely sorry. We perform all our experiments at least three times and clarify the information in legends of revised manuscript.

3. Many of the WBs are unreasonably cropped for studies investigating FBF1 SUMOylation and lack appropriate molecular mass markers, making independent interpretation as to the MW shift associated with protein SUMOylation impossible to determine. Evidence should be provided that FBF1 undergoes covalent SUMOylation and the proportion of molecules SUMOylated at an endogenous level to promote re-localization to PML-NBs.

We are sorry for this and added the molecular markers on western blots in revised manuscript.

4. A stronger rationale needs to be presented for the information described for CEP164 (page 6). It is currently unclear why this protein is even mentioned in this study. Revision of the manuscript is strongly recommended.

We are sorry for the unclear description. CEP164 is another transition fiber protein which can translocates from the ciliary base to the nucleus. It is why we tested if FBF1 behaves like CEP164 in DNA damaged cells. Our data indicated FBF1 and CEP164 play different function in different sub-compartments of the nucleus. We have revised the manuscript accordingly to give more clear background information (**Page 7** in new manuscript).

5. Data should be provided to support the statement 'Strikingly, ablation of FBF1 completely suppressed PML-NB upregulation in IR-treated IMR-90 cells' [page 7]. No evidence is present for the upregulation of PML or PML-NB constituent proteins, only for the number of bodies to change in their relative frequency per nuclei.

In Fig. 3C of revised manuscript, we show that FBF1 deficiency suppresses IR-induced PML-NB upregulation. We also performed new experiment in the **new supplementary Fig. 11B and C**, showing *FBF1* deficiency leads to the decrease of PML protein levels in IR-treated IMR-90 cells and mouse.

6. Data should be provided to support the statement ‘In agreement with the critical role of cilia in senescence regulation’ [page 7]. No data is presented to establish a critical role for cilia, only IFT88 depletion. It remains plausible that the mechanism described may be IFT88-dependent but independent of cilia or cilia biogenesis.

We thank the reviewer for the suggestion. We performed new experiments by knockdown another cilia structural component KIF3A in IMR90 cells. *KIF3A* knockdown recapitulates *IFT88* deficiency phenotypes in cilia biogenesis and senescence induced by various stressors (new **Supplementary Fig. 1**).

7. Evidence should be provided to support the statement ‘TF localization is a prerequisite for PML-NB translocation of FBF1 in stressed cells’ [page 9]. Currently there is no evidence to suggest a role in translocation, only translocation of FBF1 to PML-NBs.

We sincerely apologize for this error that leads to the confusion. “TF localization is a prerequisite for PML-NB translocation of FBF1 in stressed cells” should be “TF localization is a prerequisite for PML-NB translocation of FBF1 in stressed cells”. We correct the typo in revised manuscript (**Page 10** in new manuscript).

REVIEWERS' COMMENTS

Reviewer #1 (Remarks to the Author):

In this revised manuscript, the authors have included a significant amount of new data demonstrating that the requirement for ciliogenesis in senescence induction is a more general phenomenon that occurs in response to various senescence inducing stimuli as well as in multiple human fibroblast strains, and other cell types. The authors therefore have properly addressed my main criticism and concerns. Also my other critiques were adequately addressed. I congratulate the authors on this novel, interesting, and important study.

Reviewer #2 (Remarks to the Author):

The authors have addressed most of the criticisms and performed several of the suggested experiments to address critical points. The manuscript has been much improved and is in a nice condition now.

Reviewer #3 (Remarks to the Author):

A stress-induced cilium-to-PML-NB route drives senescence initiation

Reference: NCOMMS-22-17861A

Xiaoyu Ma, Yingyi Zhang, Yuanyuan Zhang, Xu Zhang, Yan Huang, Kai He, Chuan Chen,
Jielu Hao, Debiao Zhao, Nathan K. LeBrasseur, James L. Kirkland, Eduardo N. Chini, Qing
Wei, Kun Ling, Jinghua Hu.

Summary

This revised paper by Ma et al., proposes a role for transient cilia biogenesis in the regulation of cellular senescence and senescence-associated secretory phenotype (SASP) in response to multiple senescence-inducing stimuli (IR, H₂O₂, and IL1 β). This revised manuscript has been significantly strengthened across

multiple areas experimentally and its conclusions rephrased to avoid over interpretation. The data linking FBF1 re-localization to PML-NBs as a driver of senescence and SASP is convincing, and mechanistically enhanced by the addition of their SUMOylation studies. The authors have addressed all my previous points of concern and should be commended for the additional work presented. Minor revisions (listed below) would aid in substantiating beyond doubt the major conclusions of this study and/or improve the interpretation of the data presented within the manuscript.

Minor comments:

- 1) While the authors present data showing that FBF1 depletion fails to enhance PML expression upon IR treatment, they do not show the baseline of PML expression in FBF1 depleted cells relative to the control cells in non-IR treated cells. Thus, it is difficult for the reader to independently determine whether FBF1 depletion (directly or indirectly) influences the baseline levels of PML expression from which they measure a response to IR from (out with of the number of PML-NB foci observed per cell). The authors should quantify the transcription levels (RT-qPCR) and protein abundance (western blotting) of PML in non-IR and IR treated control and FBF1 depleted IMR-90 cells. The addition of such data would significantly strengthen the authors conclusions regarding FBF1 localization to PML-NBs in response to IR as the principle signalling mechanism.
- 2) The authors should state which statistical tests have been applied to their data in the figure legends. Currently the authors only state 'statistically analyzed'; yet described three independent statistical tests in their materials and methods section. Which test(s) were applied to the actual data presented to determine significance?
- 3) It is recommended that statistical analysis be performed for the 'mouse protection' experiments shown in fig. 6H to J between WT and Fbf1tm/tm corresponding conditions of treatment (i.e., non-IR WT vs non-IR Fbf1tm/tm and IR WT vs IR Fbf1tm/tm). This would facilitate the reader to independently interpret the strength of this data and its corresponding conclusions with respect to the use of wording 'protection'. Is there a significant difference between these groupings? While the authors acknowledge such potential differences in the discussion, there is currently no independent way to assess this statement.
- 4) The dilution for PML (sc-966) antibody used for western blotting in their study is missing from the materials and methods. This reviewer agrees that finding a PML antibody that specifically recognises mouse PML is very challenging. However, it is currently ambiguous if the PML (sc-966) antibody stated was the antibody actually used to detect mouse PML by western blotting. Similarly, there is no reference if this antibody was used to detect human PML in their mammalian cell culture experiments.

REVIEWERS' COMMENTS

We would like to express our sincere gratitude to all reviewers for their invaluable comments and constructive feedback that have greatly helped to improve the quality of our work.

Reviewer #1 (Remarks to the Author):

In this revised manuscript, the authors have included a significant amount of new data demonstrating that the requirement for ciliogenesis in senescence induction is a more general phenomenon that occurs in response to various senescence inducing stimuli as well as in multiple human fibroblast strains, and other cell types. The authors therefore have properly addressed my main criticism and concerns. Also my other critiques were adequately addressed. I congratulate the authors on this novel, interesting, and important study.

Reviewer #2 (Remarks to the Author):

The authors have addressed most of the criticisms and performed several of the suggested experiments to address critical points. The manuscript has been much improved and is in a nice condition now.

Reviewer #3 (Remarks to the Author):

A stress-induced cilium-to-PML-NB route drives senescence initiation

Reference: NCOMMS-22-17861A

Xiaoyu Ma, Yingyi Zhang, Yuanyuan Zhang, Xu Zhang, Yan Huang, Kai He, Chuan Chen, Jielu Hao, Debiao Zhao, Nathan K. LeBrasseur, James L. Kirkland, Eduardo N. Chini, Qing Wei, Kun Ling, Jinghua Hu.

Summary

This revised paper by Ma et al., proposes a role for transient cilia biogenesis in the regulation of cellular senescence and senescence-associated secretory phenotype (SASP) in response to multiple senescence-inducing stimuli (IR, H₂O₂, and IL1 β). This revised manuscript has been significantly strengthened across multiple areas experimentally and its conclusions rephrased to avoid over interpretation. The data linking FBF1 re-localization to PML-NBs as a driver of senescence and SASP is convincing, and mechanistically enhanced by the addition of their SUMOylation studies. The authors have addressed all my previous points of concern and should be commended for the additional work presented. Minor revisions (listed below) would aid in substantiating beyond doubt the major conclusions of this study and/or improve the interpretation of the data presented within the manuscript.

Minor comments:

1) While the authors present data showing that FBF1 depletion fails to enhance PML expression upon IR treatment, they do not show the baseline of PML expression in FBF1 depleted cells relative to the control cells in non-IR treated cells. Thus, it is difficult for the reader to independently

determine whether FBF1 depletion (directly or indirectly) influences the baseline levels of PML expression from which they measure a response to IR from (out with of the number of PML-NB foci observed per cell). The authors should quantify the transcription levels (RT-qPCR) and protein abundance (western blotting) of PML in non-IR and IR treated control and FBF1 depleted IMR-90 cells. The addition of such data would significantly strengthen the authors conclusions regarding FBF1 localization to PML-NBs in response to IR as the principle signalling mechanism.

As reviewer #3 suggested, we measured the transcription levels (RT-qPCR) (Fig. R. C), and redo western blotting for PML protein in non-IR and IR treated control and FBF1 depleted IMR-90 cells (Fig. R. A and B). We have replaced the western blotting data in Supplementary Figure S11b with the new one.

Fig. R. The change of PML protein and mRNA level in FBF1 knockdown IMR90 cells. A-C, Western blotting of PML (A and B) and relative mRNA levels of PML (C) in IMR-90 cells stably expressing control shRNA or shFBF1 at day 10 post-irradiation

2) The authors should state which statistical tests have been applied to their data in the figure legends. Currently the authors only state ‘statistically analyzed’; yet described three independent statistical tests in their materials and methods section. Which test(s) were applied to the actual data presented to determine significance?

We have revised the figures and legends to include the details of the statistical test used. The test results of P values also have been shown in Figures as suggested.

3) It is recommended that statistical analysis be performed for the ‘mouse protection’ experiments shown in fig. 6H to J between WT and Fbf1^{tm/tm} corresponding conditions of treatment (i.e., non-IR WT vs non-IR Fbf1^{tm/tm} and IR WT vs IR Fbf1^{tm/tm}). This would facilitate the reader to independently interpret the strength of this data and its corresponding conclusions with respect to the use of wording ‘protection’. Is there a significant difference between these groupings? While the authors acknowledge such potential differences in the discussion, there is currently no independent way to assess this statement.

We thank the reviewer for the suggestion. We have performed the statistical analysis of non-IR WT vs non-IR Fbf1^{tm/tm} and IR WT vs IR Fbf1^{tm/tm} in **Figure 6 H-J**.

4) The dilution for PML (sc-966) antibody used for western blotting in their study is missing from the materials and methods. This reviewer agrees that finding a PML antibody that specifically recognises mouse PML is very challenging. However, it is currently ambiguous if the PML (sc-966) antibody stated was the antibody actually used to detect mouse PML by western blotting. Similarly, there is no reference if this antibody was used to detect human PML in their mammalian cell culture experiments.

We apologized for missed information. We include the Cat. Number for PML antibody (sc-377390) in revised Methods segment.